# Drug-induced changes in connectivity to midbrain dopamine cells revealed by rabies monosynaptic tracing

Katrina Bartas[1], Pieter Derdeyn[1], Guilian Tian[2], Jose J Vasquez[2], Ghalia Azouz[2], Cindy M Yamamoto[2], May Hui[2], Kevin T Beier[2,3,4,5]*

[1]Program in Mathematical, Computational, and Systems Biology, University of California, Irvine, Irvine, United States; [2]Department of Physiology and Biophysics, University of California, Irvine, Irvine, United States; [3]Department of Biomedical Engineering, University of California, Irvine, Irvine, United States; [4]Department of Neurobiology and Behavior, University of California, Irvine, Irvine, United States; [5]Department of Pharmaceutical Sciences, University of California, Irvine, Irvine, United States

## eLife Assessment

This **important** study by Bartas and colleagues examined how patterns of monosynaptic input to specific cell types in the ventral tegmental area are altered by addictive drugs. The authors applied a dimensionality reduction approach (principal component analysis) and showed that various addictive drugs, and somewhat surprisingly the anesthesia alone (ketamine/xylasine), caused changes in the distribution of inputs labeled by the transsynaptic rabies virus. The evidence supporting the conclusions is overall **convincing** and provides foundational information, as well as a cautionary note on the interpretation of rabies virus-based tracing experiments.

*For correspondence:
kbeier@uci.edu

**Abstract** Addictive drugs cause long-lasting changes in connectivity from inputs onto ventral tegmental area dopamine cells (VTA[DA]) that contribute to drug-induced behavioral adaptations. However, it is not known which inputs are altered. Here, we used a rabies virus (RABV)-based mapping strategy to quantify RABV-labeled inputs to VTA cells after a single exposure to one of a variety of misused drugs – cocaine, amphetamine, methamphetamine, morphine, and nicotine – and compared the relative global input labeling across conditions. We observed that all tested addictive drugs elicited similar input changes onto VTA[DA] cells, in particular onto DA cells projecting to the lateral shell of the nucleus accumbens and amygdala. In addition, repeated administration of ketamine/xylazine to induce anesthesia induces a change in inputs to VTA[DA] cells that is similar to but different from those elicited by a single exposure to addictive drugs, suggesting that caution should be taken when using ketamine/xylazine-based anesthesia in rodents when assessing motivated behaviors. Furthermore, comparison of viral tracing data to an atlas of gene expression in the adult mouse brain showed that the basal expression patterns of several gene classes, especially calcium channels, were highly correlated with the extent of both addictive drug- or ketamine/xylazine-induced changes in RABV-labeled inputs to VTA[DA] cells. Reducing expression levels of the voltage-gated calcium channel *Cacna1e* in cells in the nucleus accumbens lateral shell reduced RABV-mediated input labeling of these cells into VTA[DA] cells. These results directly link genes controlling cellular excitability and the extent of input labeling by RABV.

**eLife digest** Dopamine is a neurotransmitter best known for its involvement in the brain's reward and motivation system, but it also has major roles in learning, habit formation, and movement. It acts as a chemical messenger that enables neurons to communicate and reinforces beneficial behaviors, often promoting reward-seeking actions.

Many drugs stimulate the brain's dopamine system by increasing dopamine release, blocking dopamine reuptake, or stimulating dopamine-producing neurons. A large proportion of dopaminergic neurons in mammals are located in the midbrain in a region known as the ventral tegmental area. However, so far, it has been unclear how drug exposure affects the inputs to these neurons across the brain.

One way to study brain circuits is to use engineered viruses as cell trackers. For example, scientists have used modified, fluorescent rabies viruses as 'neural GPS trackers' to map connections between neurons. Bartas et al. tested whether a modified rabies virus could measure how different drugs – including cocaine, amphetamine, morphine, and nicotine – affect brain-wide inputs to dopamine neurons in the ventral tegmental area.

They found that a single exposure to all addictive drugs tested induced a shared set of long-lasting changes in input neurons, many of which arise from brain regions typically associated with stress responses, such as the amygdala and related circuits. Moreover, repeated administration of a ketamine and xylazine mixture to induce anesthesia caused comparable input changes, suggesting that ketamine/xylazine anesthesia may have a similar long-term impact on the connectivity of these neurons.

These changes occurred only in select subtypes of input neurons, indicating that the effects are cell-type specific rather than global. Furthermore, the findings suggest that the rabies-based tracing reflects not only structural connectivity (i.e., the number of synapses) but also the functional state of input neurons, such as their baseline activity levels.

Bartas et al. then examined whether these effects were related to gene expression patterns in the input cells. By analyzing transcriptional data sets, they found that the expression of ion channels – in particular, calcium channels – was closely linked to drug-induced changes in the input neurons connecting with the dopamine neurons.

In summary, Bartas et al. demonstrate that addictive drugs all cause changes in similar sets of inputs to dopamine neurons that likely reflect long-term changes in input cell activity. In the future, this approach could be used to screen for experience-dependent changes in specific synapses and cell populations, further unravelling the dynamic properties of brain circuits.

## Introduction

The ventral tegmental area (VTA) contains dopamine cells (VTA[DA]) that play key roles in several behavioral processes, including reward, aversion, and motor control (*Bromberg-Martin et al., 2010*; *Lammel et al., 2014*; *Wise, 2004*). VTA[DA] cells are heterogeneous, with recent studies identifying subpopulations of VTA[DA] cells that differentially contribute to adaptive and pathological behaviors (*Kim et al., 2016*; *Lammel et al., 2011*; *Lammel et al., 2012*; *Beier et al., 2015*). Numerous studies have mapped the input–output architecture of VTA cells generally, as well as subpopulations of constituent DAergic, GABAergic, and glutamatergic cells (*Beier et al., 2015*; *Menegas et al., 2015*; *Watabe-Uchida et al., 2012*; *Beier et al., 2019*; *Cardozo Pinto et al., 2019*; *Faget et al., 2016*). These studies have helped us to understand the biased input and discrete output logic of the VTA, as well as generate hypotheses regarding the function of circuits including the VTA.

In addition to normal adaptive behaviors, VTA[DA] cells critically contribute to a variety of behaviors related to substance use disorder (SUD), including reward, sensitization, and withdrawal elicited by addictive drugs (*Kalivas and Volkow, 2005*; *Hyman et al., 2006*; *Beier et al., 2017*; *Tian et al., 2022*; *Juarez and Han, 2016*; *Mahler et al., 2019*). For a variety of misused drugs, even a single exposure triggers long-lasting changes in synaptic plasticity onto VTA[DA] cells (*Saal et al., 2003*; *Ungless et al., 2001*). These changes include both a long-lasting enhancement of excitatory synaptic strength onto VTA[DA] cells (*Ungless et al., 2001*), and development of inhibitory synaptic plasticity (*Niehaus et al., 2010*; *Bocklisch et al., 2013*). These changes are essential for a variety of drug-induced behaviors,

linking plasticity at VTA[DA] cells with downstream behavioral changes (*Bocklisch et al., 2013*; *Dong et al., 2004*; *Creed et al., 2016*). Furthermore, the changes in excitatory synaptic plasticity specifically occur onto some cell types and not others (*Lammel et al., 2011*), indicating that these adaptations are likely cell type- and pathway-specific. However, it remains unknown which sets of inputs are modified, and if these changes are consistent across misused drugs. Furthermore, synaptic plasticity is not the only type of long-lasting change that can be induced by addictive drugs. Unfortunately, even less is known about how other forms of experience-dependent changes, including elevations in input activity, may contribute to addiction-related behaviors.

We recently showed that a single dose of cocaine causes long-lasting changes in cellular activity in inputs to VTA[DA] cells (*Beier et al., 2017*; *Tian et al., 2022*). These changes occur from the globus pallidus external segment (GPe) and bed nucleus of the stria terminalis (BNST) projections to the VTA that contribute to drug-induced reward and sensitization, or withdrawal anxiety, respectively (*Beier et al., 2017*; *Tian et al., 2022*). Notably, while these studies focused on inputs from the GPe and BNST, the viral mapping results also demonstrated the existence of other input changes to the VTA that were not pursued in those studies. These unexplored connections may provide critical information about how drugs alter connectivity onto VTA[DA] cells, which in turn contribute to drug-induced behavioral changes (*Beier et al., 2017*; *Tian et al., 2022*).

In this study, we explore the brain-wide changes in RABV-labeled inputs to VTA cells elicited by a variety of different drugs, including addictive drugs, an anesthetic mixture of ketamine/xylazine (K/X), and fluoxetine, a psychoactive but non-addictive control substance. Our study provides a first-of-its-kind exploration into how even a single drug exposure can change brain-wide connectivity patterns, in addition to highlighting some concerns about the potential effects of K/X anesthesia on motivated behavior.

## Results
### Dimensionality reduction methods identify differences in inputs of VTA cell populations

We and others previously showed that DAergic, GABAergic, and glutamatergic cells in the VTA receive inputs from similar brain regions, with some quantitative differences that varied depending on the study (*Beier et al., 2019*; *Faget et al., 2016*). In our work, we also showed that connectivity differences in the VTA are largely a consequence of spatial organization of cells within the VTA (*Beier et al., 2015*; *Beier et al., 2019*; *Derdeyn et al., 2021*). The fact that the distributions of these three VTA cell populations are largely spatially overlapping may explain why the global inputs to each population are similar. However, one limitation in interpreting these studies is that they focused on individual comparisons of input counts on a per region basis. While this is a common approach in analyzing viral tracing data, and useful for both hypothesis generation and testing, it fails to account for the higher-level organization of input patterns. Our goal here is to go beyond considering single brain regions and better understand the holistic connectivity patterns of different cell populations or drug treatment conditions.

Our first goal was to explore the potential differences in input patterns generated using RABV tracing to each of the DAergic, GABAergic, and glutamatergic neuron populations in the VTA using principal component analysis (PCA). A schematic of the experiment is shown in *Figure 1A*, with representative images of brain regions of interest in *Figure 1B*. A standard representation of these data, a bar plot, is shown in *Figure 1C*. Embedding with PCA resulted in the three cell types clustering into largely distinct groups (*Figure 1D*). We considered the first two principal components (PCs), which explain the most variance in the data. In PC space, DAT-Cre brains were characterized by low PC1 and low PC2 values, GAD2-Cre brains by high PC1 values, and vGluT2-Cre brains by high PC2 values. The differences we noted were statistically significant: PC1 values of GAD2-Cre compared to DAT-Cre brains were significantly different (one-way ANOVA p = 0.010, DAT-Cre vs. GAD2-Cre multiple comparisons adjusted p = 0.0087; *Figure 1E*) and PC2 values of DAT-Cre compared to vGluT2-Cre brains were significantly different (one-way ANOVA p = 0.016, DAT-Cre vs. vGluT2-Cre multiple comparisons adjusted p = 0.016; *Figure 1F*). An advantage of PCA over other dimensionality reduction approaches is that, because it is a linear approach, the weights of each feature's contribution to each PC can be clearly interpreted. In this case, we can infer which brain regions contribute

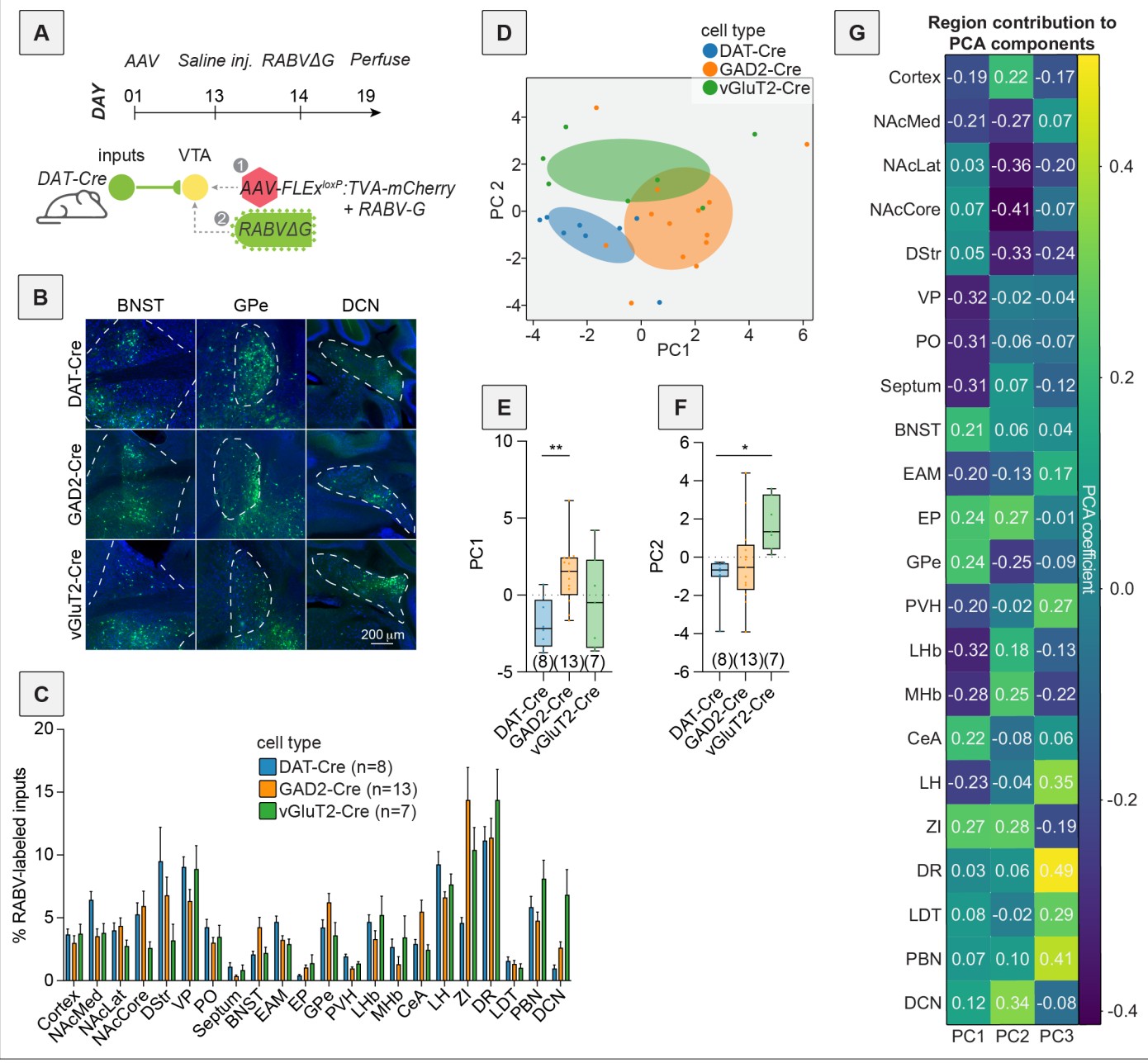

**Figure 1.** PCA-based dimensionality reduction of RABV input tracing to three different VTA cell types. (**A**) Schematic and timeline of viral injections into mice. (**B**) Representative images of the BNST, GPe, and DCN of DAT-Cre, GAD2-Cre, and vGlut2-Cre mice. Green indicates RABV-labeled cells. Scale, 200 µm. (**C**) Bar plot showing percent of RABV-labeled cells in each input region for DAT-Cre, GAD2-Cre, and vGlut2-Cre mice. (**D**) PCA of input labeling from brains from DAT-Cre, GAD2-Cre, and vGlut2-Cre mice. For this and other figures, ellipsoids were centered at the average coordinate of a condition and stretched one standard deviation along the primary and secondary axes. (**E**) Box plot comparisons of PC1. One-way ANOVA p = 0.010, pairwise t-tests DAT-Cre vs. GAD2-Cre multiple comparisons adjusted p = 0.0087, DAT-Cre vs. vGluT2-Cre p = 0.46, GAD2-Cre vs. vGluT2-Cre p = 0.19. n = 8 for DAT-Cre, n = 13 for GAD2-Cre, and n = 7 for vGluT2-Cre. (**F**) Box plot comparisons of PC2. One-way ANOVA p = 0.016, pairwise t-tests DAT-Cre vs. GAD2-Cre multiple comparisons adjusted p = 0.65, DAT-Cre vs. vGluT2-Cre p = 0.016, GAD2-Cre vs. vGluT2-Cre p = 0.051. (**G**) Heatmap of the contributions of each brain region, or feature, in the data to PCs 1–3. For this and all figures, error bars = ± 1 SEM, *p < 0.05, **p < 0.01, ***p < 0.001, ****p < 0.0001.

The online version of this article includes the following figure supplement(s) for figure 1:

**Figure supplement 1.** Representative images showing anatomical boundaries for the 22 defined input regions.

most to the positive and negative dimensions of each PC. For example, in our data, DAT-Cre and GAD2-Cre brains were separated along PC1. Brain regions that contributed to the positive dimension of PC1 were the BNST, entopeduncular nucleus (EP), GPe, central amygdala (CeA), and zona incerta (ZI) (*Figure 1G*). These inputs were enriched onto GABAergic cells relative to DA cells (*Figure 1C*). To identify where in the VTA these input populations projected, we assembled projection portraits of these inputs to the VTA as we did previously (*Derdeyn et al., 2021*). This approach uses axonal projection data available from the Allen Mouse Brain Connectivity Atlas to generate spatial heatmaps that show 'hotspots' of innervation in the midbrain (*Oh et al., 2014*). We first selected experiments where GFP was expressed in the input regions of interest. Pixel values representing projection density were pulled from a representative coronal slice containing the VTA for each region. For a given set

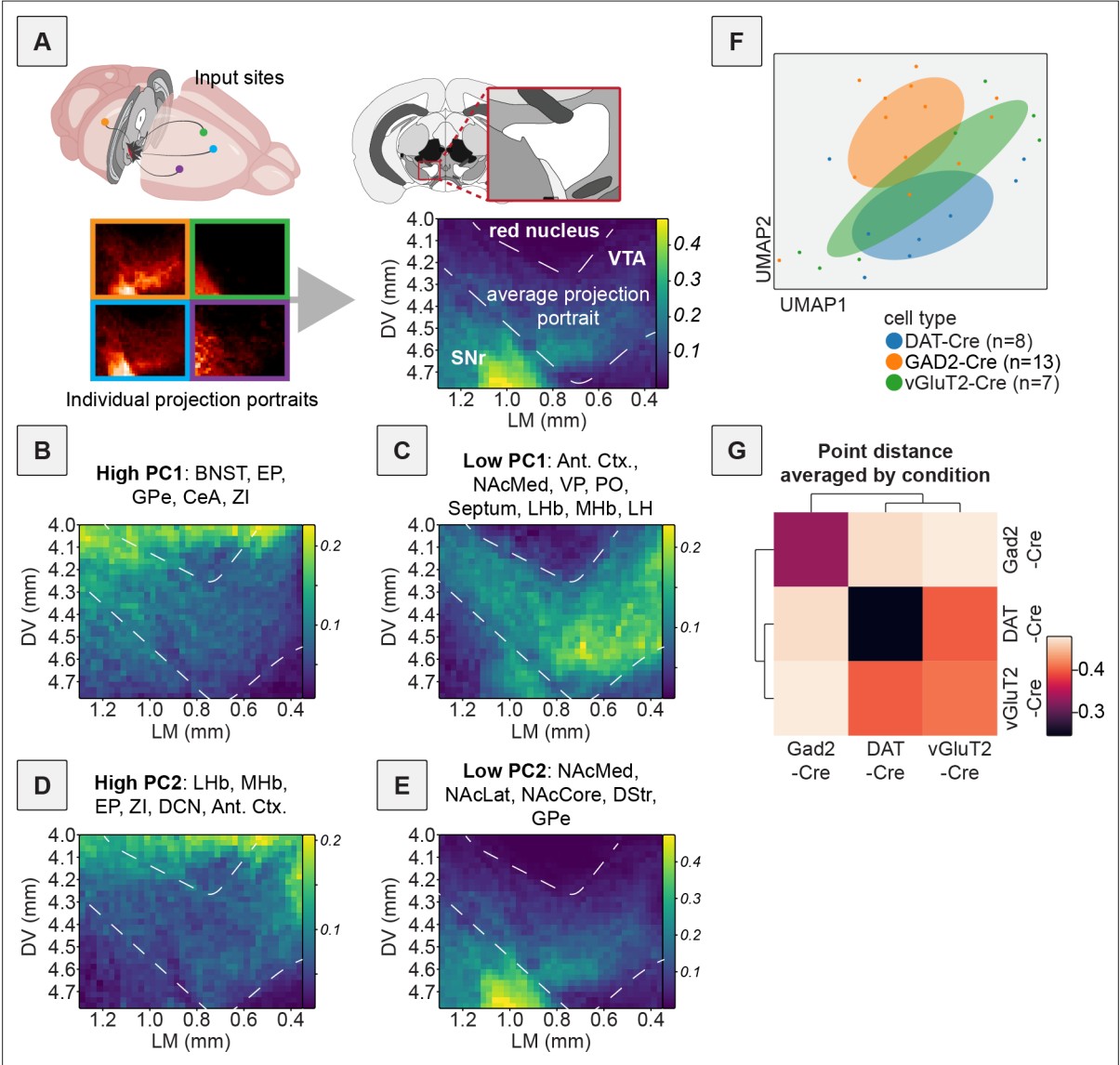

**Figure 2.** Projection portraits and UMAP of RABV inputs to VTA cells. (**A**) Schematic of how projection portraits are constructed. A representative slice of the VTA is considered for experiments from the Allen Mouse Brain Connectivity Atlas with injections into relevant brain regions that provide input to the VTA. Projection density values in this slice are averaged and visualized to obtain the projection portrait in the VTA for that group of regions. (**B**) Projection portrait of the VTA based on inputs from regions with a high positive contribution to PC1: BNST, EP, GPe, CeA, and ZI. (**C**) Projection portrait of the VTA based on inputs from regions with a large negative contribution to PC1: anterior cortex, NAcMed, VP, PO, septum, LHb, MHb, and LH. (**D**) Projection portrait of the VTA based on inputs from regions with a high positive contribution to PC2: LHb, MHb, EP, ZI, DCN, and anterior cortex. (**E**) Projection portrait of the VTA based on inputs from regions with a large negative contribution to PC2: NAcMed, NAcLat, NAcCore, DStr, and GPe. (**F**) UMAP embedding of input labeling of brains from DAT-Cre, GAD2-Cre, and vGlut2-Cre mice. (**G**) Correlogram generated using the average Euclidean distance of data points from each other from 20 UMAP embeddings.

of inputs of interest, we averaged together their projections to the VTA to see if the group had any common spatial trends (*Figure 2A*). The projection portrait of the positive PC1 inputs indicated that they preferentially project dorsally and laterally to areas surrounding the VTA (*Figure 2B*). In contrast, the regions that contributed most strongly to the negative dimension of PC1 were the anterior cortex, nucleus accumbens medial shell (NAcMed), ventral pallidum (VP), preoptic area (PO), septum, lateral habenula (LHb), medial habenula (MHb), and lateral hypothalamus (LH). These inputs were more highly labeled in DAT-Cre mice. The projection portrait of these inputs highlights the whole VTA and not surrounding regions (*Figure 2C*). While the distribution of starter cells in the VTA in DAT-Cre and GAD2-Cre mice is spatially overlapping (*Beier et al., 2015*), this analysis indicates that inputs with biases onto DA cells target the VTA itself while those biased onto GABAergic cells preferentially target areas dorsal and lateral to the VTA.

While DAT-Cre and GAD2-Cre conditions were separated along PC1, DAT-Cre and vGluT2-Cre conditions were separated along PC2. Regions that contributed to the positive dimension of PC2 included the LHb, MHb, EP, ZI, and deep cerebellar nuclei (DCN) as well as the cortex, while those contributing to the negative dimension included the NAcMed, nucleus accumbens lateral shell (NAcLat), nucleus accumbens core (NAcCore), dorsal striatum (DStr), and GPe (*Figure 2D, E*). The MHb and LHb both project medially in the VTA, whereas the NAcMed, NAcLat, NAcCore, DStr, and GPe all project laterally (*Derdeyn et al., 2021*). The projection portrait corresponding to the positive dimension of PC2 highlights the area dorsal of the VTA, as well as the medial aspect of the VTA (*Figure 2D*), while those corresponding to the negative dimension highlight the ventrolateral aspect of the ventral midbrain, including the substantia nigra pars reticulata (SNr) (*Figure 2E*). These results are consistent with the medial–lateral projection gradient that principally divides the projections into the VTA, with inputs from the basal ganglia projecting to the lateral portion of the VTA and the habenula projecting to the medial portion (*Derdeyn et al., 2021*). As inputs from the MHb and LHb contribute positive weights to the PC2 axis and vGluT2-Cre brains have the most positive PC2 values, it suggests that most of the vGluT2-expressing cells in the VTA are located within the medial portion of the VTA. This is consistent with previous reports using immunohistochemistry and in situ hybridization (*Yamaguchi et al., 2011*; *Yamaguchi et al., 2007*). In addition, the EP, ZI, and DCN all project predominantly to the red nucleus, and the cortex also innervates the red nucleus, which contains glutamatergic neurons and is located dorsal to the VTA (*Figure 2D*). While injections for our viral mapping studies targeted the VTA and most starter neurons were within the VTA, there were some starter neurons in vGluT2-Cre brains located in the red nucleus. In contrast, this was not the case in DAT-Cre brains (*Beier et al., 2019*). As these starter populations were also largely spatially overlapping, the data indicate that inputs with biases onto glutamatergic cells target the dorsomedial aspect of the VTA, while those with a relative bias onto DA cells preferentially target the ventrolateral aspect of the VTA.

While PCA is particularly useful due to the interpretability of the weights of the PCs, it cannot capture non-linear relationships in high-dimensional data, which are common in complex systems like in neuroscience. To examine the overall relationship of input patterns between DAT-Cre, GAD2-Cre, and vGluT2-Cre cell populations, we performed a second analysis on inputs from each population using Uniform Manifold Approximation and Projection (UMAP), which can capture more complex relationships than PCA. We found that DAT-Cre and GAD2-Cre brains were the most separated in UMAP space, with vGluT2-Cre brains located between them (*Figure 2F*). This is consistent with the overlap in neurotransmitters released by these cells: cells expressing DA markers such as tyrosine hydroxylase (TH) and traditional GABAergic markers such as GAD2 are nearly mutually exclusive, while there is considerable overlap between cells expressing DA markers and glutamatergic markers, as well as cells expressing GABAergic markers and glutamatergic markers (*Beier et al., 2019*; *Faget et al., 2016*; *Root et al., 2014*; *Morales and Margolis, 2017*; *Hnasko et al., 2012*). Note that while DA neurons have been found to co-transmit DA and GABA (*Kim et al., 2015*; *Tritsch et al., 2012*), this occurs through a non-canonical GABA synthesis pathway, and therefore DA cells do not express typical markers of GABAergic cells, such as GAD2, used to define the Cre line in our study.

Since individual UMAP embeddings are stochastic depending on initial seeding conditions, we examined the average Euclidean distance between pairs of brains in 20 UMAP embeddings and plotted the averaged results as a correlogram. This showed that DAT-Cre and vGluT2-Cre brains were more like one another, and the GAD2-Cre brains were the most distinct from the other conditions

(*Figure 2G*). These results are thus consistent with the known neurochemical overlap between these cells and indicate that DAT-Cre and vGluT2-Cre label more overlapping cells with each other than with GAD2-Cre. In sum, these results demonstrate that we can recapitulate the locations of different neurochemically-defined cells in the VTA with PCA and projection portraits based on information from RABV input mapping data. They also provide a validation of our approach for dissecting the relationships between input patterns and different cell types within the VTA, as the conclusions about the relationship between input regions and cell types agree with other studies using different, complementary methodologies.

## A single injection of an addictive drug changes brain-wide input patterns to VTA[DA] cells

With our dimensionality reduction approaches validated on our cell type-specific RABV data, we next applied them to examine how addictive drugs may alter VTA input patterns. We first assessed whether the psychoactive but non-addictive fluoxetine altered brain-wide input patterns relative to saline-injected mice. We found no clear differences in inputs between these conditions, which overlapped completely in PC space – with no significant differences in PC1, PC2, or PC3 between the saline and the fluoxetine brains (p > 0.05 for PC1–3) – indicating fluoxetine did not alter brain-wide input patterns to VTA[DA] cells (*Figure 3A–F*). We therefore combined these groups for additional statistical power and tested whether a single administration of one of several addictive drugs caused changes in brain-wide input patterns to VTA[DA] cells, first considered as a single group. When comparing combined controls to the drugs group, we observed a statistically significant difference in input patterns, including a reduction in inputs from the DStr and elevation in inputs from the GPe following drug injection (*Figure 3G*). In order to more fully assess which input sites may differentiate the control- vs. drug-treated groups, we again used PCA (*Figure 3H, I*). There was no significant difference between drugs and controls in PC1 (p = 0.78; *Figure 3J*). As we observed visually, the control group's PC2 and PC3 values were significantly different from the drug group's PC2 and PC3 values (p = 0.0009 and p = 0.024 for PC2 and PC3, respectively; *Figure 3K, L*). The positive axis of PC2, which was enriched in drug-treated brains, was driven by inputs from extended amygdalar areas (EAM), EP, GPe, LH, and parabrachial nucleus (PBN), while the negative axis was driven by the NAcMed, NAcCore, DStr, septum, and MHb (*Figure 3M*). Notably, the most negative weights for PC2, namely the NAcMed, NAcCore, DStr, septum, and MHb were all reduced in drug-treated brains (*Figure 3G*). Conversely, the most positive weights, namely the EAM, EP, GPe, LH, and PBN, were all increased in drug-treated brains (*Figure 3G*). Therefore, while PCA allows us to observe trends consistent with traditional methods, we can also observe the relationship between the percentage of inputs among different brains and also identify relationships that are not immediately apparent in the data when presented as bar graphs. For example, while the BNST, CeA, and ZI do not show clear differences in control vs. drug-treated brains in the bar graphs, they are all positive weights in PC2 and thus, likely change in concert with regions such as the GPe and PBN (*Figure 3G–M*).

Importantly, unlike traditional bar graph approaches, PCA allows us to isolate technical variation from the experiments, enabling us to focus more clearly on changes induced by the relevant variable, here treatment with a control substance or addictive drug. For example, PC1 does not distinguish control vs. drug-treated brains (*Figure 3H, J*), indicating this variation was inherent to the way the experiment was performed or to the biological system itself, and not related to drug or control treatment. The strongest negative weights for PC1 were the NAcLat, NAcCore, DStr, and GPe, while the strongest positive weights for PC1 were the VP, PO, LHb, and dorsal raphe (DR) (*Figure 3M*). This is consistent with connectivity differences along the medial–lateral axis of the VTA, which underlies many of the connectivity differences to different VTA[DA] subpopulations (*Beier et al., 2015*; *Beier et al., 2019*; *Derdeyn et al., 2021*). To assess whether PC1 may reflect differences in the location of starter cells in the VTA, we first assessed whether there was a systematic difference in starter cell location between conditions. We noted a high level of overlap in starter cell distribution between conditions (*Figure 4A*, *Figure 4—figure supplement 1*), indicating that this alone likely did not explain differences between control and drug-treated groups. We then assessed whether brain-to-brain variation in starter cell distribution could explain the effects observed along PC1. We colored each point in PC space according to the assessed center of injection for each experiment. We noted a clear medial–lateral gradient along PC1, but not PC2 or PC3 (*Figure 4B, C*). To test the significance

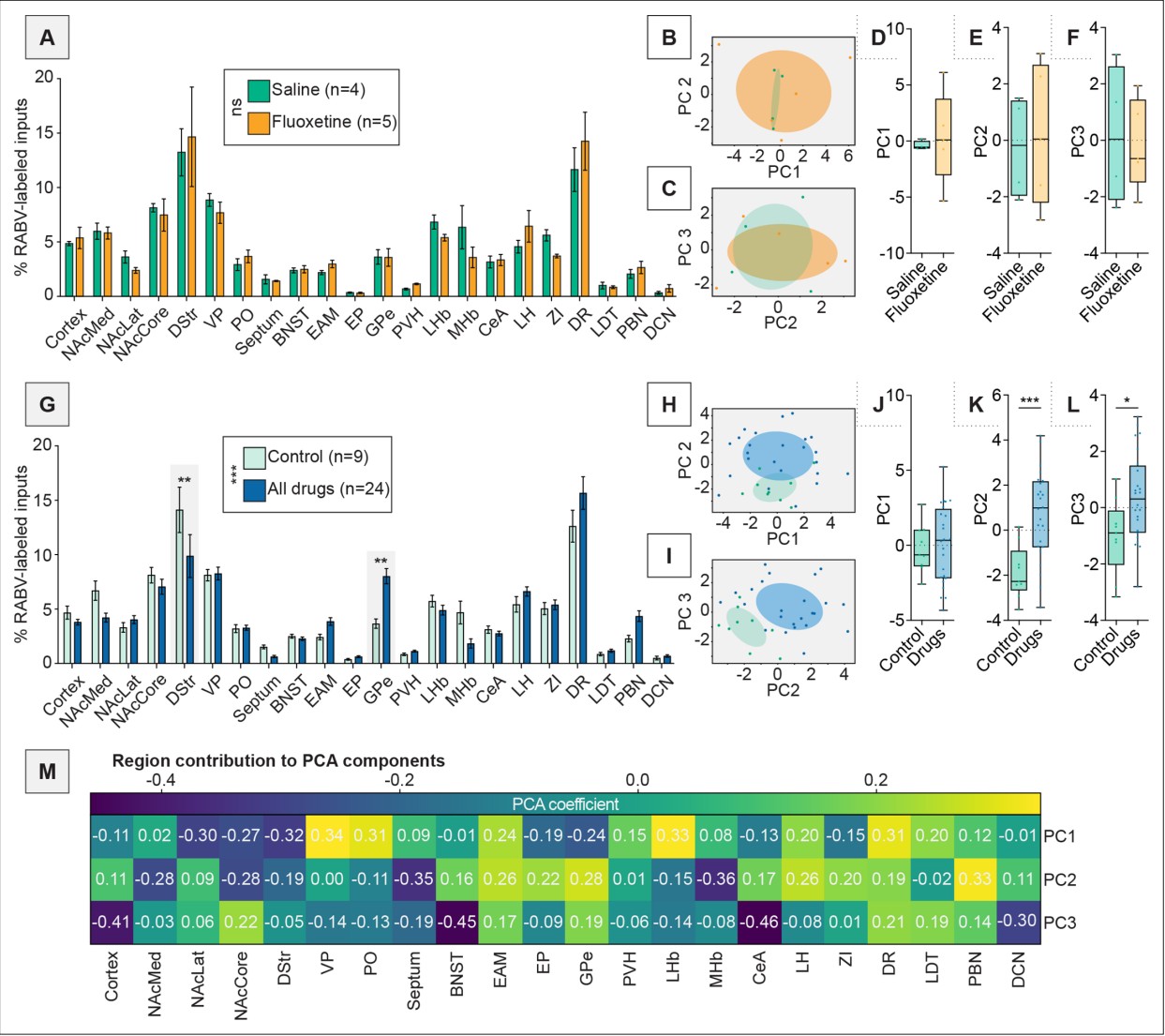

**Figure 3.** PCA can deconvolve drug-induced effects from experimental variation. (**A**) Bar graph representation of inputs from DAT-Cre mice anesthetized with isoflurane and treated with a single injection of saline or fluoxetine. n = 4 for saline, n = 5 for fluoxetine. Two-way ANOVA interaction term p = 0.9365. (**B**) PCA plot showing PC1 and PC2 of brains from saline or fluoxetine-treated mice. (**C**) PCA plot showing PC2 and PC3 of brains from saline or fluoxetine-treated mice. (**D**) Box plot of PC1, p = 0.75. n = 4 for saline, n = 5 for fluoxetine. (**E**) Box plot of PC2, p = 0.78. (**F**) Box plot of PC3, p = 0.81. (**G**) Bar graph representation of inputs from DAT-Cre mice anesthetized with isoflurane and treated with a single injection of saline or fluoxetine (controls), or one of the addictive drugs cocaine, methamphetamine, amphetamine, nicotine, or morphine. n = 9 for control, n = 24 for combined addictive drugs. Two-way ANOVA, interaction term p = 0.0001; unpaired t-tests with Sidak corrections for multiple comparisons DStr p = 0.0051, GPe p = 0.0036. (**H**) PCA plot showing PC1 and PC2 of brains from control or drug-treated mice. (**I**) PCA plot showing PC2 and PC3 of brains from control or drug-treated mice. (**J**) Box plot of PC1, p = 0.78. n = 9 for control, n = 24 for drugs. (**K**) Box plot of PC2, p = 0.0009. (**L**) Box plot of PC3, p = 0.024. (**M**) Heatmap of the contributions of each brain region to PC1–PC3 for data shown in panels H and I.

of this relationship, we performed linear regression analysis, comparing the x-coordinate of the injection center relative to location along PC1, PC2, or PC3 for each brain. We observed a highly significant positive correlation with PC1, a slightly positive but non-significant correlation with PC2, and no correlation for PC3 (*Figure 4D–F*). These results show that PC1 reflects variation in the injection site within the VTA, while PC2 reflects changes induced by addictive drugs, allowing us to isolate the changes in input connectivity induced by administration of a single addictive drug.

With our integrative methodology validated on cell type-specific RABV data and demonstrating that addictive drugs alter brain-wide input patterns to VTA[DA] cells, we applied our approaches to explore the effects of each drug specifically, as well as potential effects of the 2x use of the K/X anesthetic mixture. We first asked two questions: (1) How do injections of different addictive drugs change

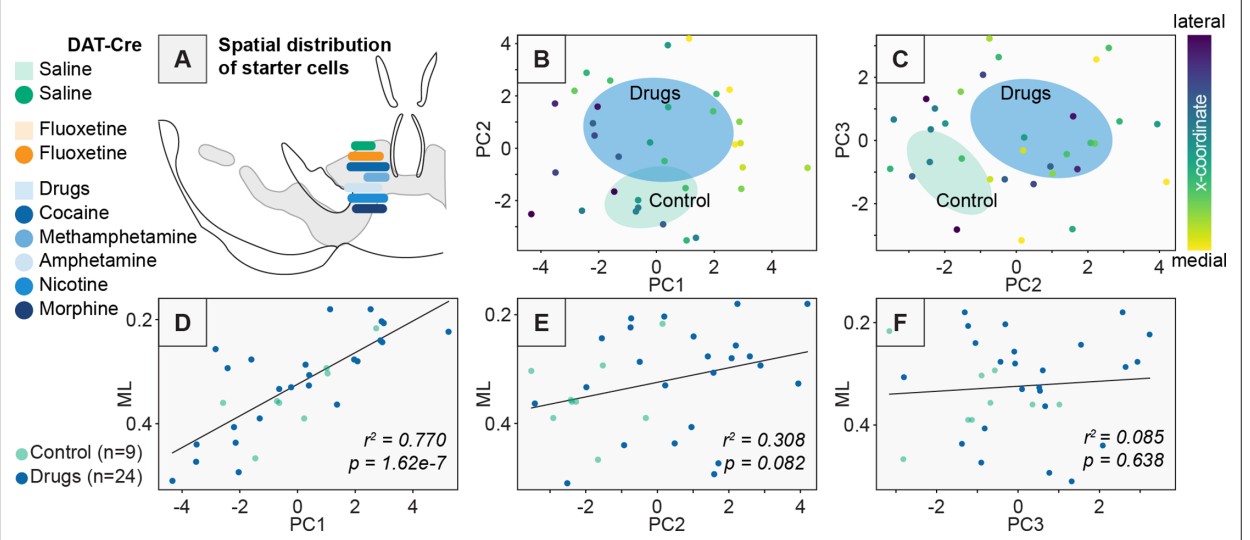

**Figure 4.** Differences in starter cell location do not explain differences in RABV input labeling patterns between addictive drug and control groups. (**A**) Distribution of the starter cell center of mass along the medial–lateral axis for each experimental condition. Saline n = 4, fluoxetine n = 5, cocaine n = 5, methamphetamine n = 4, amphetamine n = 5, nicotine n = 5, morphine n = 5. (**B**) PC1 vs. PC2 for control vs. drug conditions as shown in *Figure 3H*, with each point colored according to the medial–lateral coordinate of the center of mass of starter cells for each brain. (**C**) PC2 vs. PC3 for control vs. drug conditions as shown in *Figure 3I*, with each point colored according to the medial–lateral coordinate of the center of mass of starter cells for each brain. (**D**) Linear regression between the x-coordinate of the center of mass of starter cells for each brain vs. PC1. $r^2$ = 0.770, p = 1.62e−7. (**E**) Linear regression between the x-coordinate of the center of mass of starter cells for each brain vs. PC2. $r^2$ = 0.308, p = 0.082. (**F**) Linear regression between the x-coordinate of the center of mass of starter cells for each brain vs. PC3, $r^2$ = 0.085, p = 0.638.

The online version of this article includes the following figure supplement(s) for figure 4:

**Figure supplement 1.** Example images of starter cell distribution from mice in different treatment groups.

the connectivity patterns onto VTA cells? (2) How do these potential drug-induced differences relate to differences between unique cell types in the VTA? To answer these, we first assessed experiments with ten separate experimental conditions: two conditions of GAD2-Cre mice, and eight conditions of DAT-Cre mice. The GAD2-Cre mice were all anesthetized with isoflurane and administered either saline or cocaine one day prior to RABV injection. The DAT-Cre mice were given either isoflurane or K/X-based anesthesia. The K/X-anesthetized mice were administered saline. The isoflurane-anesthetized mice were administered either saline, fluoxetine, amphetamine, cocaine, methamphetamine, morphine, or nicotine. RABV mapping was performed from targeted VTA cells in all cases, and the brain-wide inputs were quantified. We then performed PCA on the combined RABV tracing datasets, which showed that brains from GAD2-Cre mice were in the positive dimension of PC1, while those from DAT-Cre mice were in the negative dimension (*Figure 5A*), regardless of the substance administered. This difference in PC1 was highly statistically significant (one-way ANOVA p < 0.0001; all conditions pairwise compared with GAD2-Cre, multiple comparisons adjusted p < 0.0001; *Figure 5B*). The positive dimension of PC1 received contributions from largely the same sets of regions as in the previous analysis of DAT-Cre, GAD2-Cre, and vGluT2-Cre brains, which included the BNST, EP, ZI, CeA, as well as the DCN (*Figures 1G and 5C*). Given that the same set of brain regions differentiated DAT-Cre and GAD2-Cre brains along PC1 in both datasets, this suggests that the intrinsic differences in inputs to different cell populations in the VTA are more pronounced than differences induced by addictive drugs, fluoxetine, or the method of anesthesia. Notably, these patterns were unlikely to be obtained by chance; scrambling the sample identity and percentage of RABV-labeled cells in each input site resulted in groups that overlapped in both PCA and UMAP space (*Figure 5—figure supplement 1*).

To specifically explore drug-induced differences in input labeling onto VTA^DA cells, we removed GAD2-Cre brains from the dataset. Results and representative images are shown in *Figure 5D, E*. In PCA analyses of these brains only, PC1 and PC3 largely failed to differentiate the conditions, with no significant differences observed, while PC2 significantly differentiated the conditions (one-way ANOVA p = 0.0093; saline vs. drugs multiple comparisons adjusted p = 0.034; *Figure 5F–J*). The

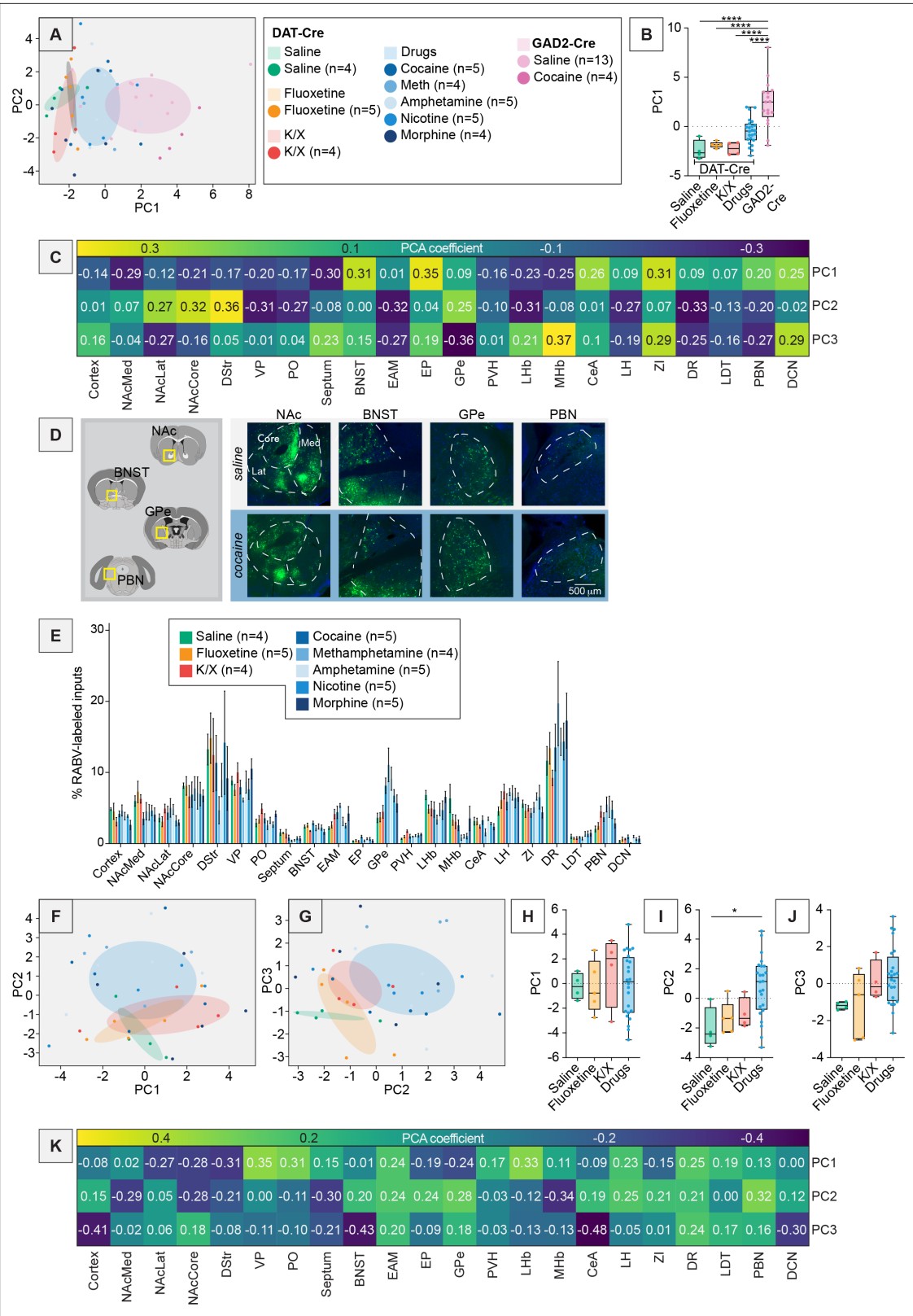

**Figure 5.** Dimensionality reduction analysis of brains from animals treated with a single drug exposure. (**A**) PCA plot showing PC1 and PC2 of brains from DAT-Cre and GAD2-Cre mice. DAT-Cre mice were anesthetized with K/X and treated with saline or anesthetized with isoflurane and treated with saline or one of the drugs cocaine, methamphetamine, amphetamine, nicotine, morphine, or fluoxetine. GAD2-Cre mice were anesthetized with isoflurane and treated with saline or cocaine. (**B**) Box plot of PC1, one-way ANOVA p < 0.0001, pairwise t-tests DAT-Cre saline vs. DAT-Cre fluoxetine

*Figure 5 continued on next page*

*Figure 5 continued*

multiple comparisons adjusted p = 0.99, DAT-Cre saline vs. DAT-Cre K/X p = 1.0, DAT-Cre saline vs. DAT-Cre drugs p = 0.23, DAT-Cre saline vs. GAD2-Cre p < 0.0001, DAT-Cre fluoxetine vs. DAT-Cre K/X p = 1.0, DAT-Cre fluoxetine vs. DAT-Cre drugs p = 0.46, DAT-Cre fluoxetine vs. GAD2-Cre p < 0.0001, DAT-Cre K/X vs. DAT-Cre drugs p = 0.31, DAT-Cre K/X vs. GAD2-Cre p < 0.0001, DAT-Cre drugs vs. GAD2-Cre p < 0.0001. n = 4 for DAT-Cre saline and DAT-Cre K/X, n = 5 for DAT-Cre fluoxetine, n = 24 for DAT-Cre drugs, and n = 17 for GAD2-Cre. (**C**) Heatmap of the contributions of each brain region to PC1–PC3 for data shown in panel A. (**D**) Representative images of brain slices from DAT-Cre mice showing, from left to right, the NAc, BNST, GPe, and PBN. Scale, 500 μm. (**E**) Bar plot showing percentage of RABV-labeled cells in each input region for the DAT-Cre mice shown in panel A. (**F**) PCA plot showing PC1 and PC2 of brains from DAT-Cre mice anesthetized with isoflurane and treated with cocaine, methamphetamine, amphetamine, nicotine, morphine, fluoxetine, or saline, or anesthetized with K/X and treated with saline. (**G**) PC2 and PC3 of the same group of brains as shown in panel F. (**H**) Box plot of PC1, one-way ANOVA p = 0.81. (**I**) Box plot of PC2, one-way ANOVA p = 0.0093, pairwise *t*-tests DAT-Cre saline vs. DAT-Cre fluoxetine multiple comparisons adjusted p = 0.94, DAT-Cre saline vs. DAT-Cre K/X p = 0.87, DAT-Cre saline vs. DAT-Cre drugs p = 0.034, DAT-Cre fluoxetine vs. DAT-Cre K/X p = 0.99, DAT-Cre fluoxetine vs. DAT-Cre drugs p = 0.10, DAT-Cre K/X vs. DAT-Cre drugs p = 0.27. (**J**) Box plot of PC3, one-way ANOVA p = 0.086. (**K**) Heatmap of the contributions of each brain region to PC1–PC3 for data shown in panels F and G.

The online version of this article includes the following figure supplement(s) for figure 5:

**Figure supplement 1.** Brains from animals treated with a single exposure to a drug (*Figure 5A*), but with the brain identities scrambled.

most negative weights to PC1 were contributed by the basal ganglia inputs (*Figure 5K*) that project to the SNr, located ventrolaterally to the VTA (*Figure 6A*), while positive contributors came from brain regions that projected more medially (LHb, MHb, dorsal raphe (DR)) and centrally to the VTA (ventral pallidum (VP), PO, EAM, paraventricular nucleus of the hypothalamus (PVH), and LH) (*Figures 5K and 6B*). This is consistent with a gradient along the medial–lateral axis (*Derdeyn et al., 2021*). As it did not differentiate the conditions, this axis likely represents experimental variations in the location of starter cell populations within the VTA (*Beier et al., 2015*; *Beier et al., 2019*; *Derdeyn et al., 2021*). The regions with the largest positive contribution to PC2 included the EAM, PBN, DR, EP, and LH, with lesser contributions from the GPe, DCN, ZI, CeA, and BNST (*Figures 5K and 6C*). These regions were over-represented in animals treated with a single administration of an addictive drug. Notably, this group of inputs contained brain regions that are involved in the brain's stress response such as the PBN, EAM, CeA, and BNST, and have been implicated in drug-induced behaviors (*Koob, 2008*; *Koob, 2009*). For example, we recently showed the importance of the GPe in conditioned place preference and locomotor sensitization and the BNST in drug-induced withdrawal (*Beier et al., 2017*;

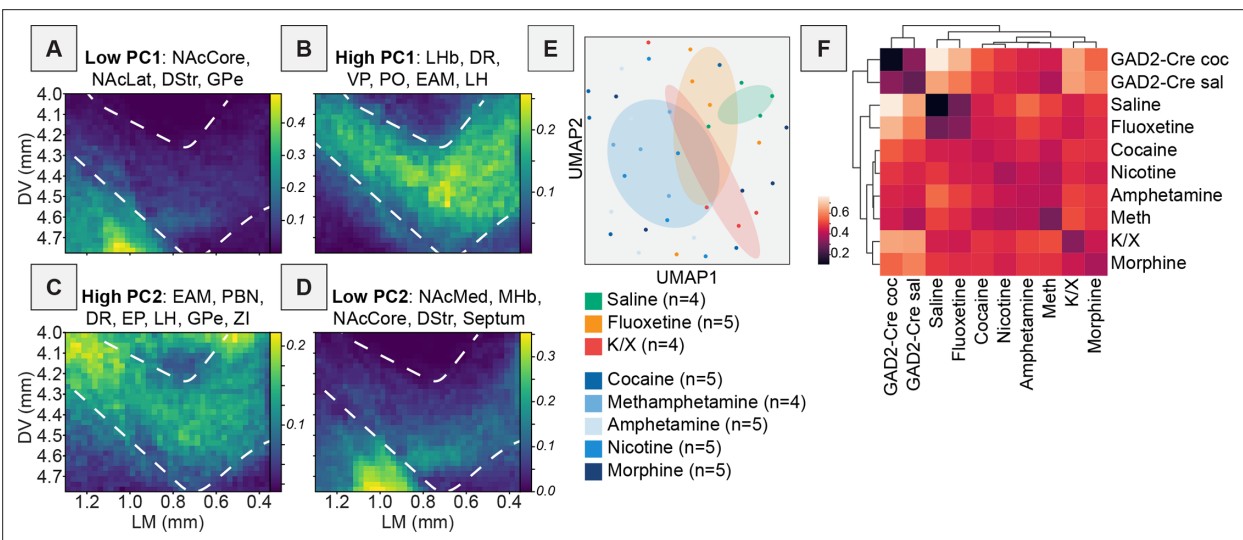

**Figure 6.** Projection portraits and UMAP plot of VTA[DA] cells following a single drug exposure. (**A**) Projection portrait of the VTA based on inputs from regions with a large negative contribution to PC1, based on data from *Figure 5K*: NAcCore, NAcLat, DStr, and GPe. (**B**) Projection portrait based on inputs from regions with a high positive contribution to PC1, based on data from *Figure 5K*: LHb, DR, VP, PO, EAM, and LH. (**C**) Projection portrait based on inputs from regions with a high positive contribution to PC2, based on data from *Figure 5K*: EAM, PBN, DR, EP, LH, GPe, and ZI. (**D**) Projection portrait based on inputs from regions with a large negative contribution to PC2, based on data from *Figure 5K*: NAcMed, NAcCore, DStr, Septum, and MHb. (**E**) UMAP embedding of data from the brains of DAT-Cre mice shown in *Figure 5E*. (**F**) Correlogram showing average Euclidean distance of data points over 20 UMAP embeddings of the data presented in panel E.

*Tian et al., 2022*). The regions with the largest negative contribution to PC2 included the NAcMed, NAcCore, DStr, Septum, and MHb (*Figures 5K and 6D*). These regions were underrepresented in animals given a single administration of an addictive drug. These results together demonstrate that addictive drugs cause long-lasting changes in input populations to DA cells, and that we can identify which sets of inputs distinguish drug- vs. control-treated brains. These changes may be functionally important, as we previously demonstrated the functional role of two inputs, the GPe and BNST, that differed between drug-treated and control conditions (*Tian et al., 2022*; *Tian et al., 2024*).

Lastly, we wanted to explore the overall relationship between input changes induced by different drugs. As before, we used UMAP to explore the higher-level organization of these conditions to compare the relationship of holistic input patterns between groups. We found that the mice anesthetized with isoflurane and treated with saline overlapped with the isoflurane-anesthetized, fluoxetine-treated group and the K/X-anesthetized and saline-treated group, but segregated from animals treated with the addictive drugs – psychostimulants, morphine, and nicotine (*Figure 6E*). Our analysis of the relationships between individual conditions, generated using the Euclidean distances between points in 20 UMAP embeddings as before, demonstrated that the expected control conditions formed a cluster (saline and fluoxetine) distinct from the other groups. Interestingly, the K/X-anesthetized animals were more closely associated with the mice treated with addictive drugs, in particular morphine, than with controls (*Figure 6F*). This suggests that addictive drugs cause similar changes in brain-wide input patterns to one another in DAT-Cre mice, and that input patterns in K/X-anesthetized animals more closely resemble morphine-treated mice than saline-treated mice. This indicates that the method of anesthesia, either using isoflurane or K/X, results in different patterns of input labeling. Given the similarity of input patterns in K/X-anesthetized mice to isoflurane-anesthetized mice treated with an addictive drug but not saline, we suspect that two doses of K/X-anesthesia as is required for RABV circuit mapping experiments triggers changes in input patterns in rodent brains that last at least the 5-day duration of these experiments.

## Anesthetic doses of K/X cause long-lasting changes in inputs to VTA$^{DA}$ → NAcLat and VTA$^{DA}$ → Amygdala cells

To examine how differences in anesthesia methods may impact brain-wide input patterns, we performed targeted comparisons. When comparing only two groups using PCA – isoflurane-anesthetized animals to K/X-anesthetized animals – these conditions completely segregated (*Figure 7A*). While the difference in PC1 was not quite significant (p = 0.086), the difference in PC2 was significant (p = 0.037). This separation was driven by positive weights from the VP, PO, EAM, PVH, LH, and PBN along PC1, which were more strongly represented in the K/X-anesthetized mice and whose combined projection portrait broadly targets the VTA (*Figure 7A–D*), and the NAcLat, DStr, GPe, and PVH along PC2, which were also more strongly represented in the K/X-anesthetized mice whose combined projection portrait targets ventrolateral to the VTA (*Figure 7A–C, E*). A similar separation of conditions was observed using UMAP (*Figure 7F*).

Next, we wanted to assess which VTA$^{DA}$ cells may be impacted by whether the animals received isoflurane or K/X-based anesthesia. To test this, we performed cell type-specific Tracing of the Relationship between Inputs and Outputs (cTRIO) for each of four different VTA$^{DA}$ subpopulations: those projecting to the NAcMed, NAcLat, Amygdala, or mPFC, and receiving K/X anesthesia and saline, isoflurane and saline, or isoflurane and cocaine. The differences in input patterns between conditions were most pronounced for VTA$^{DA}$ → NAcLat cells, where the isoflurane- and K/X-treated groups segregated (*Figure 7G, H*). Groups appeared to differ along both PC1 and PC2, though differences in PC2 were not statistically significantly different. K/X-treated brains were characterized by lower PC1 values which received contributions principally from the anterior cortex, NAcMed, NAcLat, NAcCore, septum, PVH, and MHb, as well as higher PC2 values which received contributions from the VP, PO, EAM, LHb, and LH (PC1, one-way ANOVA p = 0.0003, saline vs. cocaine multiple comparisons adjusted p = 0.011, K/X vs. cocaine multiple comparisons adjusted p = 0.0002; *Figure 7G, I–K*). The following comparisons in PC1 were statistically significant: cocaine vs. isoflurane controls (p = 0.011) and cocaine vs. K/X anesthesia (p = 0.001) (*Figure 7G*). Notably, K/X, saline-treated mice, and isoflurane, cocaine-treated mice, segregate from one another in PCA space, indicating that the K/X-induced changes are distinct from those induced by cocaine (*Figure 7G–I*). The similarity in projection portraits for weights that contribute more to the K/X-treated group in VTA$^{DA}$ → NAcLat (*Figure 7J,*

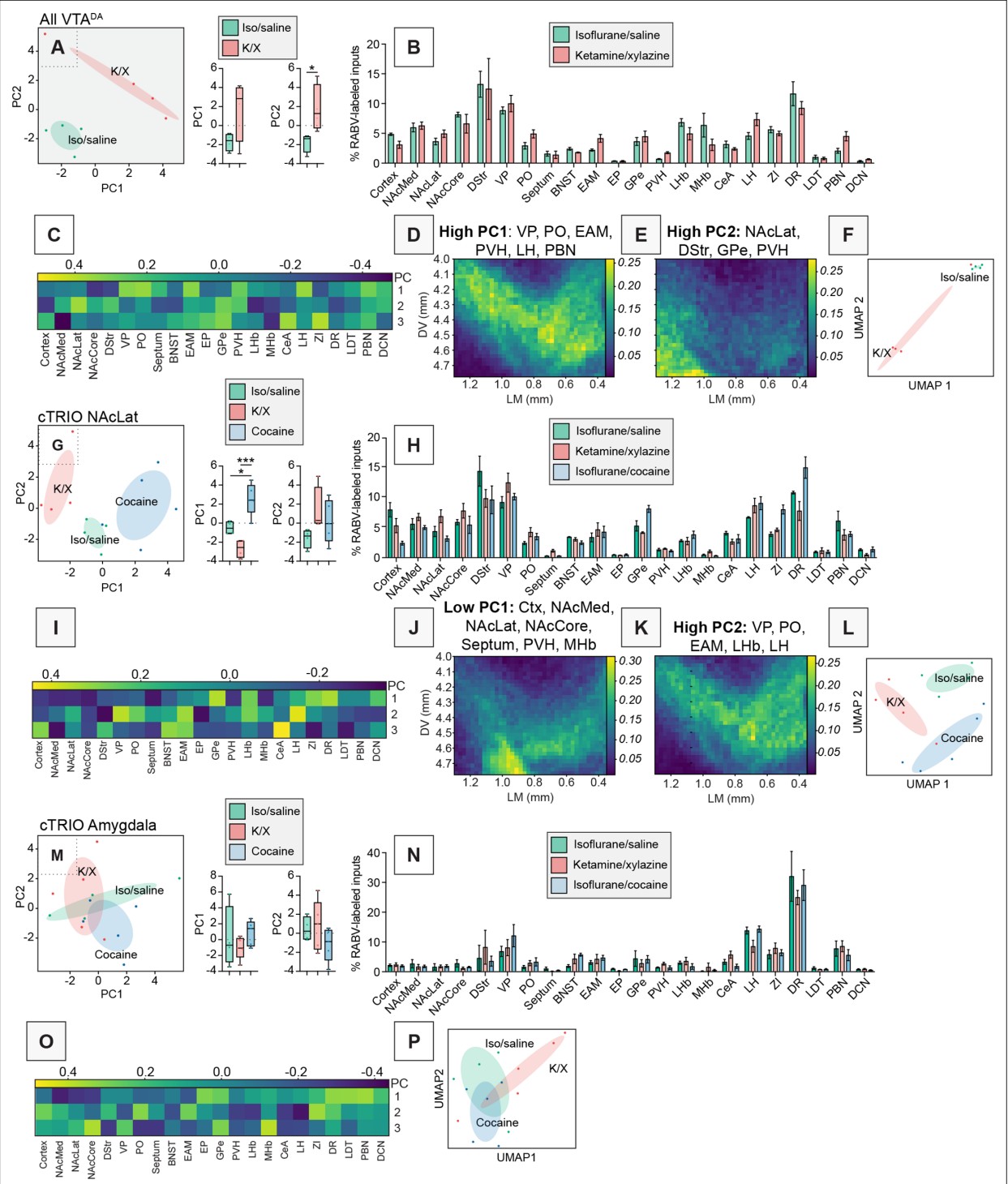

**Figure 7.** Dimensionality reduction analysis of brains from animals anesthetized with isoflurane and treated with either cocaine or saline, or anesthetized with ketamine/xylazine and treated with saline. (**A**) PCA plot of input data from VTA$^{DA}$ cells from mice anesthetized either with isoflurane or K/X and given saline. Box plot comparisons of PC1 and PC2 are shown to the right. PC1, p = 0.086, PC2, p = 0.037, unpaired *t*-tests. *n* = 4 for both. (**B**) Bar plot showing percentage of RABV-labeled cells in each input region from mice anesthetized either with isoflurane or K/X and given saline. (**C**) Heatmap of the contributions from each brain region to PC1–PC3 for data shown in panel A. (**D**) Projection portrait of regions with high PC1 contributions in all VTA$^{DA}$ cells: VP, PO, EAM, PVH, LH, and PBN. (**E**) Projection portrait of regions with high PC2 contributions in VTA$^{DA}$ cells: NAcLat, DStr, GPe, and PVH. (**F**) UMAP plot of input data from VTA$^{DA}$ cells from mice anesthetized either with isoflurane or K/X and given saline. (**G**) PCA plot of RABV input mapping experiments to VTA$^{DA}$ → NAcLat cells in mice anesthetized with isoflurane and given saline or cocaine, or anesthetized with K/X and given saline. Box plot comparisons of PC1 and PC2 are shown to the right. PC1, one-way ANOVA p = 0.0003, pairwise *t*-tests saline vs. cocaine multiple

*Figure 7 continued on next page*

*Figure 7 continued*

comparisons adjusted p = 0.011, saline vs. K/X p = 0.071, K/X vs. cocaine p = 0.0002. PC2, one-way ANOVA p = 0.16. *n* = 4 for saline and K/X, *n* = 5 for cocaine. (**H**) Bar plot showing percent of RABV-labeled cells in each input region for cTRIO experiments from VTA$^{DA}$ → NAcLat cells. (**I**) Heatmap of the contributions from each brain region to PC1–PC3 for data shown in panel G. (**J**) Projection portrait of regions with low PC1 contributions in VTA$^{DA}$ → NAcLat cells: cortex, NAcMed, NAcLat, NAcCore, septum, PVH, and MHb. (**K**) Projection portrait of regions with high PC2 contributions in VTA$^{DA}$ → NAcLat cells: VP, PO, EAM, LHb, and LH. (**L**) UMAP plot of input mapping experiments to VTA$^{DA}$ → NAcLat cells. (**M**) PCA plot of input mapping experiments to VTA$^{DA}$ → Amygdala cells. Box plot comparisons of PC1 and PC2 are shown to the right. PC1, one-way ANOVA p = 0.48. PC2, one-way ANOVA p = 0.30. *n* = 4 for saline, *n* = 5 for K/X and cocaine. (**N**) Bar plot showing percent of RABV-labeled cells in each input region for cTRIO experiments from VTA$^{DA}$ → Amygdala cells. (**O**) Heatmap of the contributions of each brain region to PC1–PC3 for data shown in panel M. (**P**) UMAP plot of input mapping experiments to VTA$^{DA}$ → Amygdala cells.

The online version of this article includes the following figure supplement(s) for figure 7:

**Figure supplement 1.** Dimensionality reduction analysis of brains from animals anesthetized with isoflurane and treated with either cocaine or saline, or anesthetized with ketamine/xylazine and treated with saline, focusing on PC3.

**Figure supplement 2.** Additional cTRIO data from mice anesthetized with K/X and given saline or isoflurane and given saline or cocaine.

**Figure supplement 3.** Inputs to brains of mice anesthetized with isoflurane and receiving saline or cocaine, or K/X-saline (*Figure 7*), but with the brain identities scrambled.

*K*) and all VTA$^{DA}$ neurons (*Figure 7D, E*) suggests that largely overlapping DA cells may be targeted for each set of experiments and is consistent with the observation that VTA$^{DA}$ → NAcLat neurons comprise the majority of VTA$^{DA}$ neurons, as noted in previous studies (*Beier et al., 2015*; *Beier et al., 2019*). As with all VTA$^{DA}$ cells, similar results were observed for VTA$^{DA}$ → NAcLat cells in both PCA and UMAP space (*Figure 7L*). These results thus serve as another proof of principle of the reliability of our approach to pull out similar results from different datasets.

In addition to VTA$^{DA}$ → NAcLat cells, we observed some segregation of isoflurane and K/X-treated conditions in VTA$^{DA}$ → Amygdala cells (*Figure 7M–P*). The lack of clear differentiation in PCA space, with only PC3 showing significant differences between the isoflurane control group and K/X anesthesia (one-way ANOVA p = 0.014, saline vs. K/X multiple comparisons adjusted p = 0.0078; *Figure 7—figure supplement 1*) makes it difficult to interpret which input sites may be contributing to the differentiation in conditions that we observed. In the VTA$^{DA}$ → NAcMed and VTA$^{DA}$ → mPFC subpopulations, the separation between isoflurane-saline, isoflurane-cocaine, and K/X-saline conditions was much less pronounced (*Figure 7—figure supplement 2*), and not significant; indeed, similar separations were observed by chance (*Figure 7—figure supplement 3*). Notably, when considering inputs to all VTA$^{DA}$ cells, the projection portraits containing the inputs most highly elevated or depressed in K/X-treated mice highlight either the entire VTA or the ventrolateral aspect of the VTA (*Figure 7D, E*). These projections are consistent with the locations of the VTA$^{DA}$ → Amygdala and VTA$^{DA}$ → NAcLat cells, respectively, within the VTA. In contrast, the portraits do not highlight the medial or ventromedial aspects of the VTA, which is the location of VTA$^{DA}$ → mPFC and VTA$^{DA}$ → NAcMed cells (*Derdeyn et al., 2021*; *Lammel et al., 2008*). These results together indicate that K/X anesthesia impacts inputs that preferentially target VTA$^{DA}$ → NAcLat or VTA$^{DA}$ → Amygdala cells.

## Exploring gene expression patterns that predict changes in RABV input labeling

We next explored if drug-induced changes in input connectivity might be related to gene expression profiles in cells located in the input sites. To do this, we calculated the correlation of gene expression data from the Allen Gene Expression Atlas (AGEA) with the difference (*Figure 8*) between RABV-labeled inputs labeled in each region in experimental conditions and isoflurane/saline-treated controls. Notably, the AGEA data are from all cells in the input regions and are not limited to those projecting to the VTA and show only baseline gene expression patterns and not those following drug administration. Nonetheless, these analyses can be a valuable starting point in associating spatial gene expression data with patterns of RABV-labeled inputs from VTA$^{DA}$ cells. After filtering AGEA data for only high-quality experiments, we were left with 4,283 genes for which we could calculate correlations within our 22 input brain regions of interest.

We first identified genes or pathways that may be linked to elevations or depressions in RABV labeling for each condition. We selected genes that were positively or negatively correlated to differences in RABV labeling between isoflurane-saline controls and (1) isoflurane-psychostimulant,

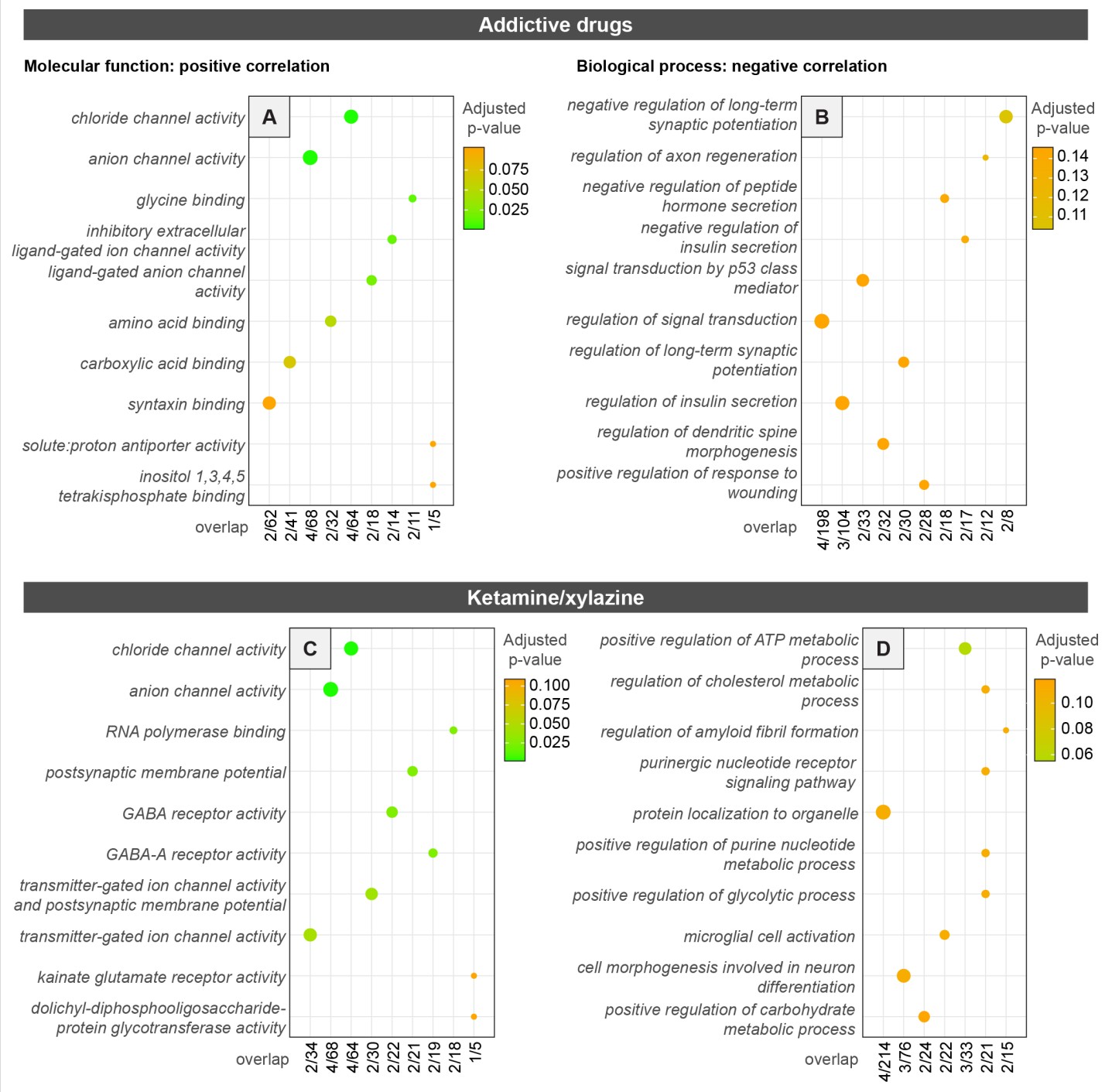

**Figure 8.** GO analysis of top 50 genes within the Allen Gene Expression Atlas correlated with differences in RABV input labeling obtained from experimental vs. control mice. (**A, B**) GO from an average of the addictive drug conditions vs. saline-treated controls, all anesthetized using isoflurane. (**C, D**) GO from K/X- vs. isoflurane-anesthetized saline-treated mice. Panels A and C show molecular function-related gene classes whose expression was positively correlated with RABV labeling differences, and panels **B and D** show GO analyses that reflect biological process-related gene classes whose expression was negatively correlated with RABV labeling differences.

The online version of this article includes the following figure supplement(s) for figure 8:

**Figure supplement 1.** GO analysis of top 50 expressed genes correlated to differences in RABV input labeling obtained from drug-treated vs. control mice.

*Figure 8 continued on next page*

*Figure 8 continued*

**Figure supplement 2.** GO analysis of top 50 expressed genes correlated to differences in RABV input labeling obtained from drug-treated vs. control mice.

**Figure supplement 3.** GO analysis of the top 50 genes correlated with differences in RABV input labeling obtained from fluoxetine vs.

morphine, nicotine, or fluoxetine, or (2) K/X-saline mice. We used the top 50 genes in the AGEA data that were either positively or negatively correlated (Spearman's $\rho$) to differences in RABV labeling and performed gene ontology (GO) analyses. Starting with addictive drugs in aggregate, the 50 most positively correlated genes (Spearman's $\rho > 0.47$, p < 0.05) were most significantly associated with inhibitory ligand-gated chloride channels (*Figure 8A*), suggesting that higher basal expression of these genes may be related to larger changes in connectivity induced by a single dose of an addictive drug. Conversely, there were many more genes that were strongly negatively correlated to drug-induced changes in RABV labeling. The 50 most negatively correlated genes (Spearman's $\rho < -0.68$, p < 0.001) were most closely associated with negative regulation of long-term synaptic potentiation (*Figure 8B*), suggesting that regulation of synaptic plasticity in input cell populations may be a key indicator of the likelihood of an addictive drug altering RABV input labeling. We repeated this analysis of the top 50 genes with each individual drug: cocaine, methamphetamine, amphetamine, morphine, and nicotine. In addition to findings in aggregate, other interesting classes of genes that were negatively associated (Spearman's $\rho < -0.50$, p < 0.05) with individual drug-induced changes in RABV labeling include regulation of synapse structural plasticity (methamphetamine), regulation of calcium ($Ca^{2+}$) ion transport (nicotine), cellular response to $Ca^{2+}$ ions (cocaine), and regulation of potassium ion transporters (nicotine) (*Figure 8—figure supplement 1*), which relate to synaptic and cellular processes of plasticity and intrinsic excitability. There were fewer than 50 genes that were significantly positively correlated, but the top 50 genes (Spearman's $\rho > 0.36$, p < 0.12) were associated with terms including anion channels (cocaine, amphetamine, and ethamphetamine) and syntaxin binding (cocaine, amphetamine, and nicotine) (*Figure 8—figure supplement 2*). We also explored which gene expression patterns may be related to input changes driven by K/X anesthesia. Anion channel activity, particularly GABA receptor expression, again shows up as one of the most significant associations to the top 50 most positively correlated genes (Spearman's $\rho > 0.37$, p < 0.10) (*Figure 8C*), while negatively correlated genes were associated with a variety of metabolic processes (*Figure 8D*). Notably, these gene expression patterns were not correlated with input changes induced by fluoxetine (*Figure 8—figure supplement 3*).

To further assess which classes of genes were the most closely related to the differences in drug-evoked RABV labeling relative to controls, we performed linear regressions of the ratios of RABV labeling in experimental vs. control conditions against the mean of the log normalized expression of AGEA RNA in situ hybridization data for a particular class of genes. We observed an overall significant and slightly negative correlation between the RABV labeling ratio and the gene expression mean ($r = -0.30$, p = $2.1 \times 10^{-3}$) for all 4,283 genes (*Figure 9—figure supplement 1A*). This genome-wide negative correlation is a useful reference to compare to correlation coefficients of subsets of genes. Among the gene expression classes that we examined, the most significant association was between the change in RABV-labeled inputs and ion channel gene expression ($r = -0.56$, p = $1.3 \times 10^{-9}$; *Figure 9A*), consistent with our results from GO analyses. The next most significant classes were neurotransmitter receptors ($r = -0.46$, p = $1.9 \times 10^{-6}$) and synapses ($r = -0.42$, p = $1.6 \times 10^{-5}$), followed by genes that have been linked to SUDs through Genome-Wide Association Studies (GWAS; $r = -0.33$, p = $7.7 \times 10^{-4}$) as well as endo/exocytosis genes ($r = -0.30$, p = $2.3 \times 10^{-3}$). Each of the above relationships showed a negative correlation of RABV ratios with gene expression (*Figure 9B, C*, *Figure 9—figure supplement 1B, C*). Except for endo/exocytosis genes, each class had a more negative, significant correlation than our genome-wide reference. We also observed that the relationships between RABV labeling ratios and gene classes that are likely irrelevant to RABV labeling, such as mitochondrial genes ($r = -0.24$, p = $1.4 \times 10^{-2}$), sugar transporters ($r = -0.26$, p = $1.0 \times 10^{-2}$), and tumor-associated genes ($r = -0.18$, p = $7.0 \times 10^{-2}$), were weaker and less significant than our classes of interest (*Figure 9—figure supplement 1D–F*).

As the class of ion channel genes was the most strongly correlated gene class to drug-induced differences in RABV-labeled input patterns, we assessed which ion channels were most closely linked to RABV labeling differences between conditions. The most significant relationship was between $Ca^{2+}$ channel genes ($r = -0.57$, p = $4.4 \times 10^{-10}$), followed by $Cl^-$ ($r = -0.39$, p = $6.1 \times 10^{-5}$), $K^+$ ($r = -0.30$,

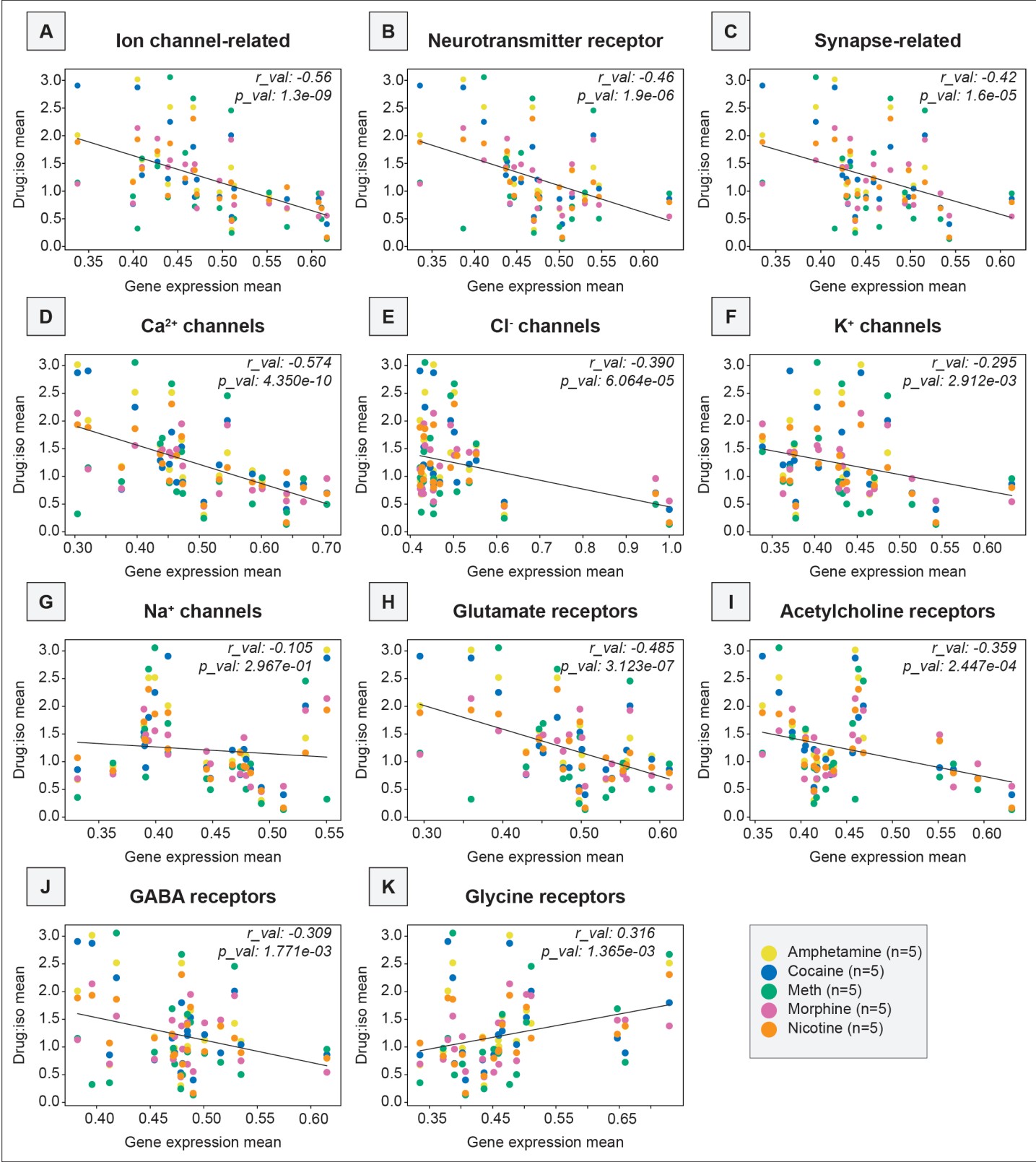

**Figure 9.** Relationship between RABV labeling ratio in addictive drug/saline-treated groups and mean basal gene expression of gene subgroups, subtypes of ion channels, and neurotransmitter receptors. Linear regressions are shown for all five examined drugs against different gene classes expressed by brain region within the Allen Gene Expression Atlas, including (**A**) ion channel-related genes, (**B**) neurotransmitter-related genes,

*Figure 9 continued on next page*

Figure 9 continued

(**C**) synapse-related genes, (**D**) Ca$^{2+}$ channels, (**E**) Cl$^-$ channels, (**F**) K$^+$ channels, (**G**) Na$^+$ channels, (**H**) glutamate receptors, (**I**) acetylcholine receptors, (**J**) GABA receptors, and (**K**) glycine receptors.

The online version of this article includes the following figure supplement(s) for figure 9:

**Figure supplement 1.** Relationship between RABV labeling ratio in addictive drug/saline-treated groups and mean basal gene expression of gene subgroups.

**Figure supplement 2.** Relationship between RABV labeling ratio in K/X-saline vs. isoflurane-saline groups and mean basal gene expression of gene subgroups.

**Figure supplement 3.** Relationship between RABV ratio labeling in fluoxetine vs. saline-treated groups, both anesthetized with isoflurane, and mean basal gene expression of gene subgroups.

p = 2.9 × 10$^{-3}$), and Na$^+$ channels (r = –0.11, p = 0.30; *Figure 9D–G*). The expression levels of these channels, in particular Ca$^{2+}$ channels, are linked to regulating a variety of cellular processes, including immediate early gene expression networks as well as cellular excitability. Ligand-gated channels (r = –0.40, p = 4.1 × 10$^{-5}$) were more strongly related with RABV labeling than voltage-gated ion channels (r = –0.33, p = 7.4 × 10$^{-4}$) or other types of ion channels (r = –0.13, p = 0.21; *Figure 9—figure supplement 1G–I*). Among the ligand-gated channels, the glutamate receptors were the most strongly correlated (r = –0.49, p = 3.1 × 10$^{-7}$; *Figure 9H*). The next most strongly correlated gene classes were acetylcholine receptors (r = –0.36, p = 2.4 × 10$^{-4}$), GABA receptors (r = –0.31, p = 1.8 × 10$^{-3}$), and glycine receptors (r = 0.32, p = 1.4 × 10$^{-3}$) (*Figure 9I–K*). Notably, glycine receptors were the only class of genes that we analyzed that was positively correlated with RABV labeling. Similar patterns were observed for K/X-anesthetized mice (*Figure 9—figure supplement 2*), but not with isoflurane-anesthetized, fluoxetine-treated mice (*Figure 9—figure supplement 3*). These results together indicate that the basal expression levels of a variety of ion channels, particularly Ca$^{2+}$ and ligand-gated ion channels, may be related to driving elevations or depressions in RABV labeling following a single injection of a variety of addictive drugs, or K/X anesthesia.

## Functional validation of link between gene expression and RABV labeling

As the correlation between RABV labeling ratios and Ca$^{2+}$ channel gene expression was negative, lower expression of Ca$^{2+}$ channel genes was linked to an elevation in drug-induced RABV input labeling. This relationship could be explained by several mechanisms. Firstly, expression of Ca$^{2+}$ channel genes could be inhibitory toward the drug-induced change in RABV transmission. Alternatively, lower Ca$^{2+}$ channel gene expression levels could indicate more potential for a drug-induced rise in gene expression level. To start differentiating between these possibilities, we performed CRISPR-based gene expression modulation of the voltage-gated calcium ion channel *Cacna1e* using CRISPRi, which reduces gene expression through blocking DNA transcription without cutting DNA (*Larson et al., 2013*; *Figure 10A, B*). *Cacna1e* was chosen as a test gene because it was one of the Ca$^{2+}$ channel genes included in *Figure 9D* that had a strong negative correlation with RABV labeling and was strongly expressed in the nucleus accumbens (*Lein et al., 2007*; *Figure 10—figure supplement 1*). Three different adeno-associated viruses (AAV) were co-injected into the NAcLat, an input site to the VTA, in DAT-Cre mice. These viruses expressed (1) Cre, (2) Cre-dependent dCas9, and (3) a Cre-dependent small guide RNA (gRNA) targeting *Cacna1e*. A two AAV mixture lacking the AAV expressing the gRNA was used as a control. The three AAV experimental mix was administered into the NAcLat in one hemisphere, while the two AAV control mix was injected into the other hemisphere. During the same surgery, AAVs expressing Cre-dependent TVA-mCherry and RABV glycoprotein were bilaterally injected into the VTA, so that RABV mapping could be done in both hemispheres (*Figure 10C, D*). Because the NAcLat, NAcCore, and GPe unilaterally project to the VTA in the same hemisphere, and not the contralateral VTA (*Beier et al., 2015*; *Beier et al., 2019*; *Beier et al., 2017*; *Tian et al., 2022*), the ratio of labeled inputs in these sites on a single hemisphere can be used to assess the relative labeling in the NAcLat in experimental and control conditions within each mouse.

We first validated that our CRISPRi strategy reduced *Cacna1e* gene expression. We designed our gRNA to cause only a mild reduction in *Cacna1e* expression to more naturally mimic potential changes in gene expression. We found that our CRISPRi perturbation reduced gene expression by

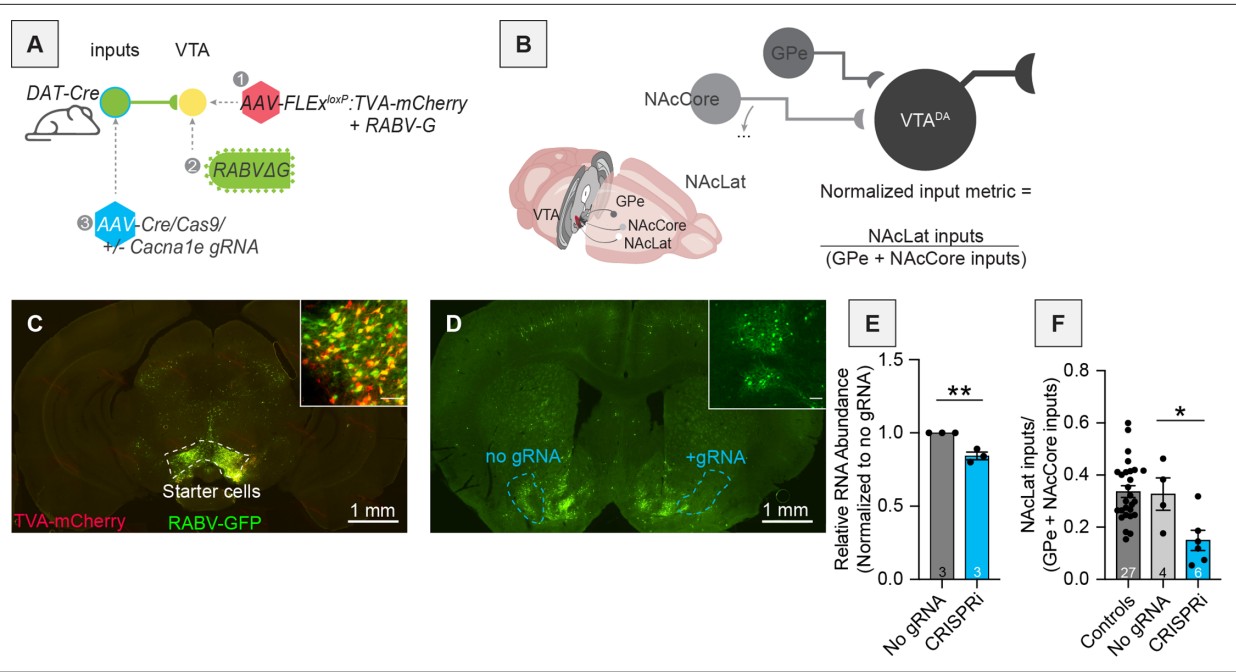

**Figure 10.** Reduction of *Cacna1e* expression in the NAcLat lowers the number of RABV-labeled inputs from the NAcLat onto VTA[DA] cells. (**A**) Schematic of viral injections performed. (**B**) Schematic of connectivity in relevant brain regions and explanation of the normalized input metric. (**C**) Representative image of fluorescence in starter cells in the VTA. Main image scale, 1 mm; inset scale, 100 µm. (**D**) Representative image of fluorescence in the NAcLat after either administration of no guide RNA (gRNA) (left) or gRNA targeting *Cacna1e* (right). Main image scale, 1 mm; inset scale, 100 µm. (**E**) qPCR results showing slight reduction of *Cacna1e* in the NAcLat. (**F**) Difference in RABV-labeled inputs after CRISPRi-mediated knockdown of *Cacna1e* compared to controls (***Beier et al., 2017***), and no gRNA. The control conditions are based on previously published data where no AAVs were injected into NAcLat (***Beier et al., 2015***).

The online version of this article includes the following figure supplement(s) for figure 10:

**Figure supplement 1.** AGEA data showing expression of *Cacna1e*.

~20% (***Figure 10E***). This knockdown of *Cacna1e* in the NAcLat via CRISPRi resulted in decreased RABV-labeled inputs from the NAcLat compared to controls where the gRNA was not included (***Figure 10F***). These results indicate that reduced levels of *Cacna1e* likely lower the number of RABV-labeled inputs from the NAcLat to the VTA and directly link the levels of *Cacna1e* expression with RABV input labeling.

## Discussion

Here we explored the long-term consequences of a single injection of a variety of addictive drugs on the connectivity of VTA[DA] cells. We first confirmed that dimensionality reduction approaches could adequately differentiate the connectivity between different cell populations in the VTA and showed which inputs differentiate DAergic, GABAergic, and glutamatergic cells in the VTA (***Figures 1 and 2***). We then showed that a variety of addictive drugs as well as K/X-based anesthesia trigger long-lasting changes in the brain-wide input patterns to VTA cells, which were clearly observable though less substantial than the differences onto different cell types or natural variation in starter cell location (***Figures 3–6***). We used our cTRIO method to identify DA cell type-specific changes induced by cocaine as well as K/X anesthesia, implicating the VTA[DA] → NAcLat and VTA[DA] → Amygdala cells as those exhibiting the most substantial input changes (***Figure 7***). We then explored intrinsic gene expression patterns throughout the brain that relate to the elevation or depression in RABV labeling by different drugs, finding that expression levels of ion channels, neurotransmitter receptors, and synapse-related genes were all significantly negatively correlated with RABV input labeling ratios, with $Ca^{2+}$ channels and glutamate receptors being the most strongly correlated (***Figures 8 and 9***). We then demonstrated the causative role of one calcium channel gene, *Cacna1e*, in influencing RABV input mapping efficiency (***Figure 10***).

## Addictive drugs induce changes in connectivity onto VTA^DA neurons from common and distributed input populations throughout the brain

Using dimensionality reduction approaches, we differentiated the input patterns to different VTA cells (*Figures 1 and 2*) and showed that these differences were larger than the differences in inputs from the same cell populations in animals given an addictive drug vs. saline (*Figure 5A*). Notably, these differences were distinct from those triggered by the psychoactive but non-addictive substance fluoxetine, as well as K/X (*Figure 5*). Differences were apparent on the second PC axis, which were driven by positive weights from the GPe, EP, EAM, LH, ZI, DR, and PBN, regions that showed elevated connectivity in drug-treated mice. Many of these regions have been implicated in cocaine addiction (*Li et al., 2021*; *López et al., 2019*; *Tan et al., 2019*; *Koob and Volkow, 2010*). While these drug-induced differences in input labeling were less substantial than intrinsic cell type connectivity differences (*Figure 5A*) or differences in starter cell location (*Figure 4*), these differences were more apparent when using dimensionality reduction methods such as PCA and UMAP. That different cell types exhibit larger differences in input patterns than the same cell type with and without drug administration is perhaps not surprising, as different populations of cells may receive different sets of inputs from one another, and these are likely more stable across experiments than the differences in connectivity triggered following a single exposure to an addictive drug. The same is likely true for spatial gradients, which we determined to be a major source of variation in RABV inputs to the VTA (*Derdeyn et al., 2021*). Notably, the minor differences in starter cell location are likely not a technical concern specific to our study, but rather inherent to all RABV mapping experiments that rely on intracranial injections, as a small amount of biological or technical variation is to be expected. The PCA approach we employ here, however, can better isolate these effects than traditional bar graph approaches and thus enable clearer observation of the effects of the relevant variable being tested, here effects of a single drug administration (or K/X anesthesia) on the input connectivity to VTA^DA cells. Importantly, these analyses may provide important insight into the circuit mechanisms underlying SUDs; for example, the GPe, LH, and PBN show up as high on PC2 from brains of mice treated with an addictive drug (*Figure 5K*). We previously showed the importance of the GPe in behaviors elicited by cocaine (*Beier et al., 2017*; *Tian et al., 2024*), the importance of the LH in reward learning and memory is well-known, in particular through neuropeptides such as orexin and melanin-concentrating hormone (*Marchant et al., 2012*; *Sheng et al., 2023*; *DiLeone et al., 2003*; *Aston-Jones et al., 2009*), and the PBN, while much less studied in the context of addiction, is involved in regulating negative emotions, pain, and aversion, and has begun to be implicated in SUD-related behaviors (*He et al., 2022*; *Pyeon et al., 2024*). Thus, our data could be used to generate new hypotheses about connections that are yet-unstudied but play important roles in addiction.

The results in this study further confirm our previous observations that a single exposure to a variety of addictive drugs triggers long-lasting changes in inputs to VTA^DA neurons. Our interpretation of these data is that the changes are largely being driven by a change in the activity of input cells, as we have shown using fiber photometry and slice electrophysiology in three separate studies (*Beier et al., 2017*; *Tian et al., 2022*; *Tian et al., 2024*). However, we have not confirmed this to be true for all cases. As a single exposure to a variety of addictive drugs alters the strength of both excitatory and inhibitory inputs in the VTA (*Lammel et al., 2011*; *Ungless et al., 2001*; *Niehaus et al., 2010*; *Bocklisch et al., 2013*), we cannot rule out that changes in synapse number or strength may also influence the observed connectivity results.

## Ketamine/xylazine anesthesia alters the input landscape of VTA^DA cells

Animals anesthetized with a K/X mixture showed a different pattern of input labeling to VTA^DA cells than animals anesthetized with isoflurane (*Figure 7A, B*). While we cannot perform RABV mapping on unanesthetized mice, making it impossible to definitively define which mode of anesthesia, or both, alters RABV input patterns, we suspect that isoflurane anesthesia has minimal effect on the brain-wide inputs to VTA^DA cells. This is largely because animals treated with an addictive drug with isoflurane anesthesia look more like those receiving K/X-saline than isoflurane-saline animals (*Figures 5 and 6*). In addition, ketamine is a drug with known addictive liability (*Liu et al., 2016*; *Ni et al., 2018*; *Dillon et al., 2003*; *Morgan et al., 2012*). While the anesthetic doses used here (100 mg/kg) are well above the doses used recreationally (~1 mg/kg), the changes that we observed with K/X anesthesia may be related to the addictive liability of ketamine. Interestingly, the K/X-induced input changes were most

similar to those induced by 10 mg/kg of morphine (*Figure 6F*). Antidepressant doses of ketamine may work in part via the opioid system; indeed, it has been known for some time that ketamine binds to the mu opioid receptor (*Ryder et al., 1978*; *Finck and Ngai, 1982*; *Hustveit et al., 1995*; *Bonaventura et al., 2021*), and it was more recently shown that the opioid antagonist naltrexone blocks the antidepressant effects of ketamine (*Williams et al., 2019*; *Williams et al., 2018*). Given that anesthetic doses of ketamine (100 mg/kg) are roughly 10 times higher than antidepressant doses of ketamine (~10 mg/kg), K/X anesthesia almost certainly engages the opioid system.

## Relationship of gene expression with the extent of drug-induced RABV input labeling

Gene expression patterns in the mouse brain are correlated with the extent of elevations and depressions of RABV labeling induced by one of several addictive drugs. We observed a slightly negative correlation of RABV labeling changes with the overall level of gene expression across the brain. We are not sure of the implications of this negative correlation, though it could suggest that if initial gene expression levels are lower, it leaves a higher potential for experience-induced elevations. We observed significant negative correlations of RABV labeling changes induced by addictive drugs with the levels of expression of ion channel genes, neurotransmitter receptor-related genes, and synapse-related genes. These results and our experiments demonstrating the functional impact of downregulating *Cacna1e* for RABV input labeling (*Figure 10*) indicate that the expression levels of these genes could be used as predictive biomarkers of which regions are likely to undergo drug-induced changes in connectivity to VTA$^{DA}$ cells. They also suggest that the expression levels of these gene classes are related to the drug-induced changes in RABV labeling.

While the AGEA is a useful resource for relating spatial gene expression information to our RABV mapping datasets, there are notable limitations. First, the AGEA dataset is not as quantitatively rigorous as modern single nucleus RNA sequencing or spatial transcriptomic datasets. Nonetheless, we believe that the overall patterns of high and low expression, especially when considered among classes of genes as was performed here, can still be informative. Furthermore, the AGEA dataset provides a static picture of gene expression levels throughout the brain, and thus does not account for potential changes in gene expression levels triggered by administration of one of the drugs used in this study. Third, the AGEA dataset considers expression levels in all cell types in a brain region, irrespective of their individual transcriptomic/molecular identities, while only a subset of cells in a region project to VTA$^{DA}$ cells. Nonetheless, our CRISPRi experiment showing that a mild reduction of *Cacna1e* reduces RABV labeling indicates the potential for causative associations between gene expression patterns and drug-induced RABV input labeling changes (*Figure 10*). This result also directly links cellular excitability and the extent to which RABV labels these cells as monosynaptic inputs to defined cell populations. While our experiments do not prove that any particular gene expression changes drive changes in RABV labeling, they provide a nice starting point for follow-up experiments.

Our RABV screen identified several input sites with elevated or depressed labeling following a single administration of an addictive drug, suggesting potential changes in excitability of these cells. Ca$^{2+}$ channel genes were also highly negatively correlated with the extent of drug-induced changes in RABV input labeling. Ca$^{2+}$ channels have been implicated in substance misuse through GWAS (*Gelernter et al., 2014*), and molecules that target these channels have shown some promise as addiction therapeutics (*Malcolm et al., 1999*; *Altamura et al., 1990*; *Rosse et al., 1994*; *Muntaner et al., 1991*; *Johnson et al., 2004*; *Vaupel et al., 1993*). It is of particular interest that Ca$^{2+}$ and not Cl$^-$, Na$^+$, or K$^+$ channels are highly correlated to RABV input labeling changes. Ca$^{2+}$ is critically involved as a secondary messenger in neurons that affects a variety of processes including long-lasting changes in transcription and synaptic transmission, whereas the other ions are mostly involved only in facilitating or inhibiting formation and propagation of the action potential. Understanding how Ca$^{2+}$ channel gene expression levels relate to drug-induced changes in the brain will help to illuminate the relationship between basal gene expression levels, drug-induced changes in RABV labeling,

and substance misuse, providing mechanistic insights into how inter-individual differences influence susceptibility to substance misuse.

# Materials and methods

## Key resources table

| Reagent type (species) or resource | Designation | Source or reference | Identifiers | Additional information |
|---|---|---|---|---|
| Gene (*Mus musculus*, males and females) | *Cacna1e* | NCBI | NM_009782.3 | |
| Strain, strain background (*Mus musculus*, males and females) | C57BL/6J | The Jackson Laboratory | Strain #: 000664, RRID:IMSR_JAX:000664 | |
| Strain, strain background (*Mus musculus*, males and females) | vGluT2-Cre | The Jackson Laboratory | Strain #: 016963, RRID:IMSR_JAX:016963 | |
| Strain, strain background (*Mus musculus*, males and females) | DAT-Cre | The Jackson Laboratory | Strain #: 006660, RRID:IMSR_JAX:006660 | |
| Strain, strain background (*Mus musculus*, males and females) | GAD2-Cre | The Jackson Laboratory | Strain #: 028867, RRID:IMSR_JAX:028867 | |
| Cell line (*Homo sapiens*) | HEK293T | ATCC | CRL-3216 RRID:CVCL_0063 | |
| Strain, strain background (*adeno-associated virus*) | AAV$_5$-CAG-FLEx$^{loxP}$-TC | University of North Carolina, vector core | | Titer: $2.4 \times 10^{12}$ genome copies (gc)/ml |
| Strain, strain background (*adeno-associated virus*) | AAV$_8$-CAG-FLEx$^{loxP}$-G | University of North Carolina, vector core | | Titer: $1.0 \times 10^{12}$ gc/ml |
| Strain, strain background (*adeno-associated virus*) | AAV$_5$-CAG-FLEx$^{FRT}$-TC | University of North Carolina, vector core | | Titer: $2.6 \times 10^{12}$ gc/ml |
| strain, strain background (*adeno-associated virus*) | AAV$_8$-CAG-FLEx$^{FRT}$-G | University of North Carolina, vector core | | Titer: $1.3 \times 10^{12}$ gc/ml |
| Strain, strain background (*canine adenovirus*) | CAV-FLEx$^{loxP}$-Flp | Plateforme de Vectorologie de Montpellier, France | | Titer: $5.0 \times 10^{12}$ gc/ml |
| Strain, strain background (*Lyssavirus rabies*) | RABVΔG GFP | Made in lab | | Titer: $5.0 \times 10^{8}$ colony forming units (cfu)/ml |
| Strain, strain background (*adeno-associated virus*) | AAV$_9$-CamKII-0.4-Cre-SV40 | Addgene | 05558-AAV9 | Titer: $1.5 \times 10^{13}$ gc/ml |
| Recombinant DNA reagent | pJEP317 pAAV-U6SaCas9gRNA(SapI)-EFS-GFP-KASH-pA | Addgene | Plasmid #113694, RRID:Addgene_113694 | |
| Recombinant DNA reagent | pJEP317-pAAV-U6-Cacna1eAC-EFS-GFP-KASH-pA | Made in lab | | |
| Recombinant DNA reagent | pHelper | A gift from Matthew Banghart at the University of California, San Diego | | |
| Recombinant DNA reagent | pAAV2-8 | Addgene | Plasmid #112864, RRID:Addgene_112864 | |
| Recombinant DNA reagent | pAAV2-5 | Addgene | Plasmid #232922, RRID:Addgene_232922 | |
| Commercial assay or kit | Q5 Site-directed mutagenesis kit | New England Biolabs | E0554S | |
| Commercial assay or kit | Luna Universal One-Step RT-qPCR Kit | New England Biolabs | E3005L | |
| Chemical compound, drug | PEIMAX | Polysciences | 24765-100 | |

*Continued on next page*

*Continued*

| Reagent type (species) or resource | Designation | Source or reference | Identifiers | Additional information |
|---|---|---|---|---|
| Chemical compound, drug | PEG 8000 | Fisher Scientific | AA4344336 | |
| Chemical compound, drug | DAPI | Thermo Fisher Scientific | D1306 | |
| Chemical compound, drug | Isoflurane | Patterson Veterinary | 78938441 | |
| Chemical compound, drug | Ketamine | Patterson Veterinary | 78935790 | |
| Chemical compound, drug | Xylazine | Patterson Veterinary | 78945244 | |
| Chemical compound, drug | Vetameg | Patterson Veterinary | 78924375 | |
| Chemical compound, drug | Vetbond | Patterson Veterinary | 78055031 | |
| Chemical compound, drug | Fluoxetine | Tocris Bioscience | 0927 | |
| Chemical compound, drug | Cocaine | Sigma | C5776-1G | |
| Chemical compound, drug | Morphine | Patterson Veterinary | 78924699 | |
| Chemical compound, drug | Methamphetamine | Cayman Chemical company | 13997 | |
| Chemical compound, drug | Amphetamine | Sigma | A5880-1G | |
| Chemical compound, drug | Nicotine | Tocris Bioscience | 3546 | |
| Software, algorithm | Python | | Version 3.11 | |
| Other | Superfrost Plus microscope slides | Fisher Scientific | 1255015 | |
| Other | Coverslips | Thermo Fisher Scientific | 152460 | |
| Other | Fluoromount-G | SouthernBiotech | 0100-01 | |
| Other | TRIzol | Thermo Fisher Scientific | 15596026 | |

## Experimental procedures

### Animals

Mice were housed on a 12-hr light–dark cycle with food and water ad libitum. Males and females from a C57BL/6J background were used for all experiments in approximately equal proportions. Mice were approximately 3–4 months of age at the time of experiments. All surgeries were done under isoflurane or ketamine/xylazine anesthesia. All procedures complied with the animal care standards set forth by the National Institute of Health and were approved by the University of California, Irvine's Institutional Animal Care and Use Committee (IACUC; protocol number AUP-21-125) and Institutional Biosafety Committee (BUA-R261).

### Stereotaxic surgery

Mice were anesthetized with 3–4% isoflurane and maintained during surgery at 1–1.5% isoflurane, or with a mixture of ketamine (100 mg/ml) and xylazine (5 mg/ml). Mice were secured in a stereotaxic apparatus (Stoelting). Under aseptic conditions, guide holes were drilled, and viruses were infused into the target sites using a glass capillary attached to a microinjection pump (WPI, UMP3T). 100 nl of AAVs or 500 nl of EnvA-pseudotyped RABV were infused at a rate of 1.6 nl/s. The glass capillary remained in place for 5 min following the infusion to allow for virus diffusion. After infusion, the surgical incision sites were closed with either sutures or Vetbond tissue adhesive (Patterson Veterinary). Vetameg (flunixin, Patterson veterinary) was administered for pain management and topical bacitracin was applied to prevent infection at the incision site.

For viral mapping experiments from a defined cell type, 100 nl of a 1:1 volume mixture of AAV$_5$-CAG-FLEx$^{loxP}$-TCB and AAV$_8$-CAG-FLEx$^{loxP}$-RABV-G was injected into the VTA of DAT-Cre (*Bäckman et al., 2006*), GAD2-Cre (*Taniguchi et al., 2011*), or vGluT2-Cre mice (*Vong et al., 2011*). Thirteen days later, animals were given a single injection of an addictive drug, fluoxetine, or saline. The following

day, 500 nl of EnvA-pseudotyped RABV was injected into the VTA. Animals were sacrificed five days later. For cTRIO experiments shown in *Figure 7*, during the first surgery, AAV$_5$-CAG-FLEx$^{FRT}$-TCB and AAV$_8$-CAG-FLEx$^{FRT}$-RABV-G were injected into the VTA, and CAV2-FLEx$^{loxP}$-Flp was injected into an output site (NAcLat, Amygdala, NAcMed, mPFC) during the same surgery. All other procedures were the same as above. RABV-labeled cells throughout the brain were manually quantified, as done previously (*Beier et al., 2015*; *Beier et al., 2019*; *Beier et al., 2017*).

The injection coordinates and volumes used were as follows:

VTA: 100 nl AAV and 500 nl RABV, AP –3.2, LM 0.4, DV –4.2
NAcMed: 250 nl, AP +1.78, LM 0.4, DV –4.1
NAcLat: 250 nl, AP +1.45, LM 1.75, DV –4.0
Amygdala: 500 nl, AP –1.43, LM 2.5, DV –4.5
mPFC: 1 µl into mPFC two injections of 500 nl, one at AP +2.15, LM 0.27, DV –2.1 and another at AP +2.15, LM 0.27, DV –1.6

## Drug administration

Cocaine was administered (intraperitoneal injections) at a dose of 15 mg/kg, morphine at 10 mg/kg, methamphetamine at 2 mg/kg, amphetamine at 10 mg/kg, nicotine at 0.5 mg/kg, and fluoxetine at 10 mg/kg.

## Quantification of RABV data

RABV mapping data were analyzed as described previously (*Beier et al., 2015*; *Beier et al., 2019*). All RABV mapping data included were previously published (*Beier et al., 2015*; *Beier et al., 2019*; *Beier et al., 2017*), except for input tracing from mice treated with a single dose of methamphetamine. The SAD B19 vaccine strain was used for all experiments. A total of 118 samples were included in this study (*Supplementary file 1*). Exclusion criteria included mice in which fewer than 200 input cells were labeled, or in cases where injections were mistargeted (e.g., starter cells in the substantia nigra pars compacta). Experimenters were blinded to the experimental condition when data quantification was being performed. For starter cell center of mass calculations, we mapped the distribution of starter neurons, as defined by TVA-mCherry expression along the A-P axis, and in the sections at the center point along this axis, defined the center of labeling along the M–L axis.

## CRISPRi experiments

### Plasmid construction

The CRISPRi system was adapted from Kumar's method (*Kumar et al., 2018*). pJEP317 pAAV-U6SaCas9gRNA(SapI)-EFS-GFP-KASH-pA was obtained from Addgene and was used as a template to clone pJEP317-pAAV-U6-Cacna1eAC-EFS-GFP-KASH-pA using the Q5 Site-directed mutagenesis kit (primers: 5'-acctccgtaaGTTTTAGTACTCTGGAAACAG-3' and 5'-atgaacccttGGTGTTTCGTCCTTTCCAC-3') to insert the gRNA sequence (AAGGGTTCATACCTCCGTAA) for *Cacna1e*. The gRNA sequence was selected from the IDT Predesigned Alt-R CRISPR-Cas9 guide RNA resource (here). The plasmid for CRISPRi (pJEP309-pAAV-EFS-dSaCas9-KRAB-Dio-pA), pAAV2-8, and pAAV2-5 were obtained from Addgene. pHelper plasmid was a gift from Matthew Banghart at the University of California, San Diego.

### AAV production, purification, and titering

Generation of AAVs from plasmids and AAV purification by Iodixanol gradient ultracentrifugation were carried out as described on Addgene's website with minor modifications. Briefly, pHelper, pAAV2-5/pAAV2-8, and pJEP309/pJEP317-pAAV-U6-Cacna1eAC-EFS-GFP-KASH-pA plasmids were transfected into HEK293T cells using PEIMAX (Polysciences, cat# 24765-100). The molar ratio for pHelper:pAAV2-5:transfer plasmid was 1:1:1, and the ratio of total plasmids to PEIMAX was 1:3. Seventy-two hours post-transfection, cell lysate and medium precipitated with PEG were purified by iodixanol gradient. Purified AAVs were titered using primers against viral ITRs (5'-GGAACCCCTAGTGATGGAGTT-3' and 5'-CGGCCTCAGTGAGCGA-3'). Cell lines are routinely tested for mycoplasma contamination using a standard PCR detection protocol (*Uphoff and Drexler, 2011*). HEK 293T cells tested negative for mycoplasma contamination; no further validation was performed.

## Viral injection

Prior to surgery, DAT-Cre mice were anesthetized with isoflurane (4% induction, 1–1.5% maintenance) and secured on a stereotaxic frame for precise localization of injection sites. For assessment of in vivo genome editing, two conditions were generated. For the experimental condition, 600 nl of a 1:1:1 volume mix of AAV$_8$-EFS-dSaCas9-KRAB-Dio, AAV$_9$-CamKII-0.4-Cre-SV40 (Addgene, cat# 05558-AAV9) and AAV$_5$-U6-Cacna1eAC-EFS-GFP-KASH was injected into the NAcLat. For a no gRNA control condition, 400 nl of a 1:1 volume mix of AAV$_8$-EFS-dSaCas9-KRAB-Dio and AAV$_9$-CamKII-0.4-Cre-SV40 was infused into the NAcLat.

For mapping inputs to VTA cells in mice with *Cacna1e* downregulation in NAcLat, during a single surgery, 300 nl of a of a 1:1 volume mix of AAV$_5$-FLEx$^{loxP}$-TC and AAV$_5$-FLEx$^{loxP}$-RABV-G was injected into the VTA bilaterally, 600 nl of a 1:1:1 volume mix of AAV$_8$-EFS-dSaCas9-KRAB-Dio, AAV$_9$-CamKII-0.4-Cre-SV40, and AAV$_5$-U6-Cacna1eAC-EFS-GFP-KASH was injected into the left NAcLat, and 400 nl of a 1:1 volume mix of AAV$_8$-EFS-dSaCas9-KRAB-Dio and AAV$_9$-CamKII-0.4-Cre-SV40 was injected into the right NAcLat. This study design enabled controls within the same mice to prevent systematic biases due to differences in treatment of the mice. After 14 days, RABVΔG was injected into the VTA. Mice were sacrificed five days later, and brains processed as described below.

## Brain section preparation, imaging, and cell counting

Five days after RABV injection, mice underwent transcardial perfusion with 1x PBS followed by 4% formaldehyde in 1x PBS. Brains were extracted and post-fixed in 4% formaldehyde for 24 hr, and then dehydrated with 30% sucrose in 1x PBS. Brains were cut into 60 µm slices using a cryostat and mounted on Superfrost Plus microscope slides (Fisher, cat # 1255015). Once the brain sections were fully dried on the slide, they were rehydrated in 1xPBS containing DAPI at a dilution of 1:1000 (Thermo Fisher, cat# D1306). After 10 min, the liquid was removed, and Fluoromount-G (SouthernBiotech, cat# 0100-01) was applied to the sections. Coverslips (Thermo Scientific, cat# 152460) were used to cover the brain sections on the slides. Images of the entire slides were captured using an Olympus IX83 microscope, and manual cell counting was performed on the acquired images.

## Determination of CRISPRi-mediated knockdown efficiency

To determine the efficacy of CRISPRi knockdown of *Cacna1e* expression levels, 10 days following AAV injections into the NAcLat, NAcLat brain tissue was dissected out from the brains of injected mice, and tissue was pooled (5 mice for *Cacna1e* gRNA and 3 mice for no gRNA control). Total RNA was extracted with TRIzol, and 50 ng of total RNA was used for RT-qPCR using the Luna Universal One-Step RT-qPCR Kit (New England Biolabs, cat# E3005L) following the manufacturer's instructions. Phosphoglycerate Kinase 1 (PGK1) was used as the housekeeping gene for normalization of *Cacna1e* mRNA levels.

## Dimensionality reduction of RABV input data

For RABV input mapping experiments, quantified input cells were binned into 22 regions: Cortex (anterior cortex), NAcMed, NAcLat, NAcCore, DStr, VP, PO, Septum, BNST, EAM, EP, GPe, PVH, LHb, MHb, CeA, LH, ZI, DR, LDT, PBN, and DCN. Altogether, this dataset consists of the input cell counts for these 22 regions across 118 mouse brains, with *n* = 4 or 5 for each condition.

RABV input tracing data were reported as the percentage of total quantified labeled RABV inputs, as reported previously (*Beier et al., 2015*; *Beier et al., 2019*). These data were further scaled using *Z*-score normalization. PCA was then implemented on these normalized data using the scikit-learn library in Python with default parameters (*Pedregosa, 2011*). Weights for PCA models were visualized as heatmaps, reflecting the contribution of each feature to each PC. To test for differences between groups visualized in PC space, we assessed the value for each brain for each PC and if comparing 2 conditions, used an unpaired *t*-test, and if >2 conditions, a one-way ANOVA and if p < 0.05, followed by pairwise t-tests with multiple comparisons corrections (Dunnett or Tukey, as appropriate). Input values were the first three PCs, and significant differences between conditions for each PC (drugs vs. controls, for example) were denoted as having a p-value of less than 0.05. Results are reported where relevant in the text or in *Supplementary file 2*.

UMAP, implemented using the UMAP library in Python, was used to generate alternative visualizations of the RABV input data (*McInnes et al., 2018*). To choose and standardize our UMAP parameters

across analyses, we investigated which methods placed mouse brains of the same type and treatment close together most consistently. For the distance metric parameter, the metrics we found to perform best were Euclidean and cosine, but we chose to use the Euclidean metric for consistency with previous analyses (*Derdeyn et al., 2021*). For the number of neighbors parameter, we chose a large number, ⅛ the size of each of our datasets, for a more global view of the data. Other parameters were kept at the default values. To account for the stochasticity of individual embeddings, we applied UMAP 20 times to each data set and found the results to be very similar, as found in previous analyses (*Derdeyn et al., 2021*). The results were then presented as correlograms. Clustered heatmaps and plots of these data were made using a combination of the Python packages seaborn (*Waskom, 2021*) and matplotlib (*Hunter, 2007*). The seaborn clustermap function was implemented with default values, including Euclidean distance as the distance metric.

Each PCA and UMAP representation of data shows each animal as a point, and each point is colored by the relevant condition, either genotype or drug administered. To better visualize the distribution of a given condition in PCA and UMAP space, we overlaid the points with ellipsoids. Ellipsoids were centered at the average coordinate of a condition and stretched one standard deviation along the primary and secondary axes.

## Brain region selection in the Allen Mouse Brain Atlas

To compare our data with connectivity data from the Allen Institute, we converted our 22 brain regions into regions or groups of regions defined by the Allen Atlas. This was necessary as our region annotations defined while doing cell counts of the RABV data were at times different from the Allen Mouse Brain Atlas-defined regions. Twenty-two brain regions were used for RABV analyses, but these were reduced to 20 brain regions (NAcMed, NAcLat, and NAcCore were merged into the NAc) for comparison with Allen Atlas data. In several cases, multiple Allen-defined regions were used to constitute a single region that we defined, as in the case of the cortex. A summary of these conversions is as follows:

| Full brain region name | Standard abbreviation | Allen abbreviations |
|---|---|---|
| Cortex | Anterior cortex | PL, ILA, ORB, GU, AI, MOs, MOp |
| Nucleus accumbens (medial) | NAcMed | ACB |
| Nucleus accumbens (lateral) | NAcLat | ACB |
| Nucleus accumbens (core) | NAcCore | ACB |
| Dorsal striatum | DStr | STRd |
| Ventral pallidum | VP | PALv |
| Preoptic area | PO | MPO, LPO |
| Septum | Septum | LSX, MS |
| Bed nucleus of the stria terminalis | BNST | BST |
| Extended amygdala area | EAM | MEA, AAA, NLOT, COA, MA, BMA, EPv, NDB |
| Internal globus pallidus | EP | GPi |
| External globus pallidus | GPe | GPe |
| Paraventricular nucleus of the hypothalamus | PVH | PVH |
| Lateral habenula | LHb | LH |
| Medial habenula | MHb | MH |
| Central nucleus of the amygdala | CeA | CEA |
| Lateral hypothalamus | LH | LHA |
| Zona incerta | ZI | ZI |

*Continued on next page*

*Continued*

| Full brain region name | Standard abbreviation | Allen abbreviations |
|---|---|---|
| Dorsal raphe nucleus | DR | DR, PAG |
| Laterodorsal tegmental nucleus | LDT | LDT |
| Parabrachial nucleus | PBN | PB |
| Deep cerebellar nuclei | DCN | CBN |

It is important to note that these conversions are approximate, and brain-region specific results should be interpreted with caution, as there are known differences between atlases used in this work (*Chon et al., 2019*). Representative images with boundaries showing every region on at least one section are shown in *Figure 1—figure supplement 1*.

## Allen Mouse Brain Connectivity Atlas projection portrait construction

Projection portraits of the VTA were made using data from the Allen Mouse Brain Connectivity Atlas (*Oh et al., 2014*), as described previously (*Derdeyn et al., 2021*). Briefly, data from 20 input brain regions, as defined in the above table, were downloaded and used for analyses. Cre-based and non-Cre-based experiments representative of each brain region were chosen for visualization. Labeled cell bodies at the site of injection often spread into adjacent brain regions or only covered a part of a region. Several experiments were chosen to account for this variation and give a better estimate for each region's projections. Experiment IDs are recorded in *Supplementary file 1* and in our previous work (*Derdeyn et al., 2021*). Connectivity of brain regions of interest to the VTA was visualized by showing a representative slice of the VTA and coloring the image based on high or low connectivity according to averaged AAV tracing data. AAV tracing data were averaged across experiments and across brain regions that grouped together according to weights assigned by PCA, as shown in *Figures 2, 6, and 7*.

Data for projection portraits and for subsequent network construction was accessed using the python Allen SDK package, developed and maintained by the Allen Institute for Brain Science.

## Gene expression data access and processing

Gene expression data was downloaded from the Allen Institute's Anatomic Gene Expression Atlas (AGEA) (*Lein et al., 2007*), an in situ hybridization dataset which has annotated expression of 19,932 genes to every brain region defined in the Allen Mouse Brain Atlas. We used code written by Eli Cornblath (https://github.com/ejcorn/mouse_abi_tool, *ejcorn, 2021*) to access and download gene expression data for our regions of interest, as done in a previous study (*Brynildsen et al., 2020*). To ensure only high-quality data were included in our analyses, we filtered experiments based on metrics used in previous analyses of this dataset (*Fulcher et al., 2019*). We retained genes for which there were data from coronal sections, or where the Pearson correlation coefficient across experiments was over 0.5 for a particular gene in our regions of interest. This filtering left us with 4,283 unique genes for the 36 Allen-defined brain regions to be aggregated to compare with our 22 quantified input sites from the RABV mapping data. We averaged data among brain regions that were defined differently in the atlas vs. our RABV quantifications, which resulted in a total of 20 brain regions for comparative analyses, as detailed above. Data were normalized using a scaled sigmoid function, a method that is robust to outliers and was used in previous analyses of these data (*Sian et al., 1999*).

## Gene expression data analysis and comparison to RABV data

To investigate how basal gene expression might be correlated to connectivity changes after drug exposure, we used processed data from the AGEA. For each experimental condition, the RABV data for the drug-treated animals were compared to those of the saline-treated controls. To quantify this comparison, we computed a ratio, as used in *Figure 9* and *Figure 9—figure supplements 1–3*, by dividing the average RABV counts in each brain region for the experimental condition by the average RABV counts in each brain region for the saline condition. This resulted in a vector of 22 ratios for each experimental condition, one ratio for each brain region. We also computed a difference, as used in *Figure 8* and *Figure 8—figure supplements 1–3*, by subtracting the average RABV counts in each

brain region for the experimental condition by the average RABV counts in each brain region for the saline control condition. This resulted in a vector of 22 differences for each drug comparison, one difference for each brain region.

We calculated the correlation of the condition-control differences with gene expression data for each region. The Spearman correlation coefficient between the gene expression across regions and our RABV differences was used because the correlative relationship between these values cannot be assumed to be linear. A similar method was used in previous analyses (*Fulcher et al., 2019*). A high positive correlation between a gene and an experimental condition ratio was interpreted as a greater likelihood of a brain region to exhibit RABV labeling changes after drug exposure if this gene is highly expressed before drug exposure. A negative correlation between a gene and an experimental condition ratio was taken to mean this region would be more likely to exhibit RABV labeling changes after drug exposure if the gene expression level is low.

We performed GO analysis on the top 50 most highly correlated genes and on the top 50 most negatively correlated genes. GO analysis was performed using the EnrichR package (*Xie et al., 2021*). Lists of genes with highly positive or negative correlations to our condition differences were compared to GO annotations using the coding language R. The 2021 version of two databases was queried: GO Biological Process and GO Molecular Function. GO enrichment analysis results were visualized in dot plots using the package ggplot2, as shown in *Figure 8* and *Figure 8—figure supplements 1–3*. The 'overlap' term on the x-axis of these plots means the number of genes in the correlated gene lists that were also within the list of genes for the corresponding GO term.

We next compared the drug/isoflurane ratios for a group of addictive drugs to gene expression of subsets of genes of interest (*Figure 9*, *Figure 9—figure supplement 1*). Similar analyses were done for ketamine/xylazine and fluoxetine (*Figure 9—figure supplements 2 and 3*). The subgroups of genes used were synapse-related genes (*Ji et al., 2020*), ion channel-related gene (*Wang et al., 2015*), neurotransmitter receptor-related genes (*Berchtold et al., 2013*), differentially expressed genes related to substance dependence GWAS (*Shi et al., 2021*), and exo- and endocytosis-related genes (*Saheki and De Camilli, 2012*). Other subsets of genes of interest were defined using the Guide to Pharmacology database (*Harding et al., 2022*). Correlation coefficients (Spearman's $\rho$) and p-values are reported in the figures.

## Acknowledgements

We would like to thank Hemmings Wu and Zhoule Zhu for their thoughtful critique of our manuscript. This work was supported by the NIH (R00 DA041445, DP2 AG067666, R01 DA054374, R01 DA056599, and R01 NS130044), Tobacco Related Disease Research Program (T31KT1437 and T31P1426), American Parkinson Disease Association (APDA-5589562), Alzheimer's Association (AARG-NTF-20-685694), New Vision Research (CCAD2020-002), Brightfocus Foundation (A2022031S), and the Brain and Behavior Research Foundation (NARSAD 26845) to KTB, NIH T3 2GM008620 and NIH F30 DA056215 to MH, T32 GM136624 to KB and PD, and NSF GRFP DGE-1839285 to PD.

## Additional information

### Competing interests
Katrina Bartas: is currently employed at Zoetis, Inc (the work was conducted prior to this employment, and this employment is unrelated to the subject matter of the manuscript). The other authors declare that no competing interests exist.

### Funding

| Funder | Grant reference number | Author |
|---|---|---|
| National Institutes of Health | R00 DA041445 | Kevin T Beier |
| National Institutes of Health | DP2 AG067666 | Kevin T Beier |

| Funder | Grant reference number | Author |
| --- | --- | --- |
| National Institutes of Health | R01 DA054374 | Kevin T Beier |
| National Institutes of Health | R01 DA056599 | Kevin T Beier |
| National Institutes of Health | R01 NS130044 | Kevin T Beier |
| Tobacco-Related Disease Research Program | T31KT1437 | Kevin T Beier |
| Tobacco-Related Disease Research Program | T31P1426 | Kevin T Beier |
| American Parkinson Disease Association | APDA-5589562 | Kevin T Beier |
| Alzheimer's Association | AARG-NTF-20-685694 | Kevin T Beier |
| New Vision Research | CCAD2020-002 | Kevin T Beier |
| BrightFocus Foundation | A2022031S | Kevin T Beier |
| Brain and Behavior Research Foundation | NARSAD 26845 | Kevin T Beier |
| National Institutes of Health | T32 GM008620 | May Hui |
| National Institutes of Health | F30 DA056215 | May Hui |
| National Institutes of Health | T32 GM136624 | Katrina Bartas Pieter Derdeyn |
| National Science Foundation | GRFP DGE-1839285 | Pieter Derdeyn |

The funders had no role in study design, data collection, and interpretation, or the decision to submit the work for publication.

## Author contributions

Katrina Bartas, Data curation, Formal analysis, Performed computational analyses of the data; Pieter Derdeyn, Data curation, Formal analysis, Generated network constructions and projection portraits; Guilian Tian, Investigation, Performed CRISPRi experiments; Jose J Vasquez, Investigation, Performed CRISPRi experiments; Ghalia Azouz, Investigation, Performed CRISPRi experiments; Cindy M Yamamoto, Methodology; May Hui, Visualization, Prepared figures; Kevin T Beier, Conceptualization, Resources, Data curation, Formal analysis, Supervision, Funding acquisition, Investigation, Visualization, Methodology, Writing – original draft, Project administration, Writing – review and editing, Performed majority of experiments and wrote manuscript

## Author ORCIDs

Pieter Derdeyn https://orcid.org/0000-0002-1545-5572
Jose J Vasquez https://orcid.org/0000-0002-4922-1520
May Hui https://orcid.org/0000-0002-6231-7383
Kevin T Beier https://orcid.org/0000-0002-4934-1338

## Ethics

All procedures complied with the animal care standards set forth by the National Institute of Health and were approved by the University of California, Irvine's Institutional Animal Care and Use Committee (IACUC; protocol number AUP-21-125) and Institutional Biosafety Committee (BUA-R261).

Reviewer #1 (Public review): https://doi.org/10.7554/eLife.93664.3.sa1
Reviewer #2 (Public review): https://doi.org/10.7554/eLife.93664.3.sa2
Reviewer #3 (Public review): https://doi.org/10.7554/eLife.93664.3.sa3
Author response https://doi.org/10.7554/eLife.93664.3.sa4

# Additional files

## Supplementary files

MDAR checklist

Supplementary file 1. List of experimental conditions included in this manuscript for brain-wide RABV tracing analysis.

Supplementary file 2. Table of statistical comparisons performed in this manuscript.

## Data availability

The current manuscript is largely a computational study, so no data have been generated for most of this manuscript, except for *Figure 10*; raw data are available for *Figure 10* on Dryad at https://doi.org/10.5061/dryad.gxd25481q. Code used for this work is available on GitHub (https://github.com/ktbartas/Bartas_et_al_eLife_2024 copy archived at *Bartas and Derdeyn, 2026*).

The following dataset was generated:

| Author(s) | Year | Dataset title | Dataset URL | Database and Identifier |
|---|---|---|---|---|
| Bartas K, Derdeyn P, Beier KT | 2026 | Drug-induced changes in connectivity to midbrain dopamine cells revealed by rabies monosynaptic tracing | http://dx.doi.org/10.5061/dryad.gxd25481q | Dryad Digital Repository, 10.5061/dryad.gxd25481q |

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
