## [Editor Report · eLife Assessment]

This **important** study by Bartas and colleagues examined how patterns of monosynaptic input to specific cell types in the ventral tegmental area are altered by addictive drugs. The authors applied a dimensionality reduction approach (principal component analysis) and showed that various addictive drugs, and somewhat surprisingly the anesthesia alone (ketamine/xylasine), caused changes in the distribution of inputs labeled by the transsynaptic rabies virus. The evidence supporting the conclusions is overall **convincing** and provides foundational information, as well as a cautionary note on the interpretation of rabies virus-based tracing experiments.

---

## [Referee Report · Reviewer #1 (Public review)]

Summary:

In this study, the authors mapped afferent inputs to distinct cell populations in the ventral tegmental area (VTA) using dimensionality reduction techniques, revealing markedly different connectivity patterns under normal versus drug-treated conditions. They further showed that drug-induced changes in inputs were negatively correlated with the expression of ion channels and proteins involved in synaptic transmission. Functional validation demonstrated that knockdown of a specific voltage-gated calcium channel led to reduced afferent inputs, highlighting a causal link between gene expression and connectivity.

The authors have clearly addressed the reviewers' previous comments. The study's earlier weaknesses were thoroughly discussed, and additional data were provided to strengthen the findings. Overall, the revised version incorporates more extensive datasets and analyses, resulting in a more robust and compelling study.

---

## [Referee Report · Reviewer #2 (Public review)]

The application of rabies virus (RabV)-mediated transsynaptic tracing has been widely utilized for mapping cell-type-specific neural connectivities and examining potential modifications in response to biological phenomena or pharmacological interventions. Despite the predominant focus of studies on quantifying and analyzing labeling patterns within individual brain regions based on labeling abundance, such an approach may inadvertently overlook systemic alterations. There exists a considerable opportunity to integrate RabV tracing data with the global connectivity patterns and the transcriptomic signatures of labeled brain regions. In the present study, the authors take an important step towards achieving these objectives.

Specifically, the authors conducted an intensive reanalysis of a previously generated large dataset of RabV tracing to the ventral tegmental area (VTA) using dimension reduction methods such as PCA and UMPA. This reaffirmed the authors's earlier conclusion that different cell types in the VTA, namely dopamine neurons (DA) and GABAergic neurons, exhibit quantitatively distinct input patterns, and a single dose of addictive drugs, such as cocaine and morphine, induced altered labeling patterns. Additionally, the authors illustrate that distinct axes of PCA can discriminate experimental variations, such as minor differences in the injection site of viral tracers, from bona fide alterations in labeling patterns caused by drugs of abuse. While the specific mechanisms underlying altered labeling in most brain regions remain unclear, whether involving synaptic strength, synaptic numbers, pre-synaptic activities, or other factors, the present study underscores the efficacy of an informatics approach in extracting more comprehensive information from the RabV-based circuit mapping data.

Moreover, the authors showcased the utility of their previously devised bulk gene expression patterns inferred by the Allen Gene Expression Atlas (AGEA) and "projection portrait" derived from bulk axon mapping data sourced from the Allen Mouse Brain Connectivity Atlas. The utilization of such bulk data rests upon several limitations. For instance, the collection of axon mapping data involves an arbitrary selection of both cell type-specific and non-specific data, which might overlook crucial presynaptic partners, and often includes contamination from neighboring undesired brain regions. Concerns arise regarding the quantitativeness of AGEA, which may also include the potential oversight of key presynaptic partners. Nevertheless, the authors conscientiously acknowledged these potential limitations associated with the dataset.

Notably, building on the observation of a positive correlation between the basal expression levels of Ca2+ channels and the extent of drug-induced changes in RabV labeling patterns, the authors conducted a CRISPRi-based knockdown of a single Ca2+ channel gene. This intervention resulted in a reduction of RabV labeling, supporting that the observed gene expression patterns have causality in RabV labeling efficiency. While a more nuanced discussion is necessary for interpreting this result (see below), overall I commend the authors for their efforts to leverage the existing dataset in a more meaningful way. This endeavor has the potential to contribute significantly to our understanding of the mechanisms underlying alterations in RabV labeling induced by drugs of abuse.

Finally, drawing upon the aforementioned reanalysis of previous data, the authors underscored that a single administration of ketamine/xylazine anesthesia could induce enduring modifications in RabV labeling patterns for VTA DA neurons, specifically those projecting to the nucleus accumbens and amygdala. Given the potential impact of such alterations on motivational behaviors at a broader level, I fully agree that prudent consideration is warranted when employing ketamine/xylazine for the investigation of motivational behaviors in mice.

Comments on revisions:

In the re-revised version, the authors have addressed all of my previous comments. I no longer have any major concerns.

---

## [Referee Report · Reviewer #3 (Public review)]

Summary:

Authors mapped monosynaptic inputs to dopamine, GABA, and glutamate neurons in the ventral tegmental area (VTA) under different anesthesia methods, and under drug (cocaine, morphine, methamphetamine, amphetamine, nicotine, fluoxetine). First, they propose an analysis method to separate the actual manipulation effects from the variability caused by experimental procedures. Using this method, they found differences in the anatomical location of monosynaptic inputs to dopamine neurons under different conditions, and identified some key brain areas for such separation. They also searched the database for gene expression patterns that are common across input brain areas, with some changes by anesthesia or drug administration.

Strengths:

The whole-brain approach to address drug effects is appealing, and their conclusion is clear. The methodology and motivation are clearly explained.

Weaknesses:

While gene expression analyses may not be related to their findings on the anatomical effects of drugs, this is a nice starting point for follow-up studies.

---

## [Author Response]

The following is the authors’ response to the original reviews.

**Public Reviews:**

**Reviewer #1(Public review):**
Summary:In this study, the authors distinguished afferent inputs to different cell populations in the VTA using dimensionality reduction approaches and found significantly distinct patterns between normal and drug treatment conditions. They also demonstrated negative correlations of the inputs induced by drugs with gene expression of ion channels or proteins involved in synaptic transmission and demonstrated the knockdown of one of the voltage-gated calcium ion channels caused decreased inputs.Weaknesses:(1) For quantifications of brain regions in this study, boundaries were based on the Franklin-Paxinos (FP) atlas according to previous studies (Beier KT et al 2015, Beier KT et al 2019). It has been reported significant discrepancies exist between the anatomical labels on the FP atlas and the Allen Brain Atlas (ref: Chon U et al., Nat Commun 2019). Although a summary of conversion is provided as a sheet, the authors need to describe how consistent or different the brain boundaries they defined in the manuscript with Allen Brain Atlas by adding histology images. Also, I wonder how reliable the annotations were for over a hundred of animals with manual quantification. The authors should briefly explain it rather than citing previous studies in the Material and Methods Section.

We thank the reviewer for attention to this point; indeed, neuroanatomical detail is often overlooked in modern neuroscience, occasionally leading to spurious conclusions. We acknowledge that there are significant discrepancies in brain region definitions across atlases, which can make cross-study comparisons difficult. Here, all cells were manually quantified by Dr. Kevin Beier, as in previous studies (Beier et al., Cell 2015; Nature 2017; Cell Reports 2019; Tian et al., Cell Reports 2022; Tian et al., Neuron 2024; Hubbard et al., Neuropsychopharmacology, 2025). As such, these studies are internally consistent as relates to the definition of brain regions, which is critical here since our analysis in this manuscript relates to data quantified only by a single individual. Several brain regions were quite easy to distinguish anatomically, such as the medial habenula and lateral habenula. Others, such as the extended amygdala area, are much more difficult. We have now provided example images in Figure S1 that detail the anatomical boundaries that we used, overlayed on images of Neurotrace blue (fluorescent Nissl stain).

(2) Regarding the ellipsoids in the PC, although it's written in the manuscript that "Ellipsoids were centered at the average coordinate of a condition and stretched one standard deviation along the primary and secondary axes", it's intuitively hard to understand in some figures such as Figure 2O, P and Figure S1. The authors need to make their data analysis methods more accessible by providing source code to the public.

The source code is now available to the public at https://github.com/ktbartas/Bartas_et_al_eLife_2024, which is noted in the Code Availability statement. The code for generating ellipsoids is in the first notebook, `0-dataexploration-master-euclidean.ipynb`, in the function `confidence_ellipse`, which is called from `make_pca_plots` and `umap_and_heatmap`. Example plots are all live in the notebooks as can be viewed directly from GitHub.

(3) In histology images (Figure 1B and 3K), the authors need to add dashed lines or arrows to guide the reader's attention.

Dashed lines have been added to these figure panels as requested.

(4) In Figure 2A and G, apparently there are significant differences in other brain regions such as NAcMed or PBN. If they are also statistically significant, the authors should note them as well and draw asterisks(*).

We appreciate the care in ensuring that statistics are being applied and shown appropriately. In panel A (now Figure 3A), the Two-way ANOVA interaction term was not significant (p = 0.9365), we did not find it justified to do further comparisons. However, for Figure 3G, the interaction term was significant (p = 0.0001), and thus further pairwise comparisons were performed with Sidak's correction for multiple comparisons. When done, the only two brain regions that were significantly different were the DStr (p = 0.0051) and GPe (p = 0.0036). While the NAcMed and PBN visually look different, according to the corrected statistics, they were not significantly different (NAcMed p = 0.5037, PBN p = 0.8123). The notations in our original figure thus accurately reflected these statistics.

(5) In Figure 2N about the spatial distribution of starter cells, the authors need to add histology images for each experimental condition (i.e. saline, fluoxetine, cocaine, methamphetamine, amphetamine, nicotine, and morphine) as supplement figures

We have now provided these as Figure S2.

(6) In the manuscript, it is necessary to explain why Cacna1e was selected among other calcium ion channels.

We have added a sentence to the "Functional validation of link between gene expression and RABV labeling" section (lines 722-724).

**Reviewer #2 (Public review):**
The application of rabies virus (RabV)-mediated transsynaptic tracing has been widely utilized for mapping celltype-specific neural connectivities and examining potential modifications in response to biological phenomena or pharmacological interventions. Despite the predominant focus of studies on quantifying and analyzing labeling patterns within individual brain regions based on labeling abundance, such an approach may inadvertently overlook systemic alterations. There exists a considerable opportunity to integrate RabV tracing data with the global connectivity patterns and the transcriptomic signatures of labeled brain regions. In the present study, the authors take an important step towards achieving these objectives. Specifically, the authors conducted an intensive reanalysis of a previously generated large dataset of RabV tracing to the ventral tegmental area (VTA) using dimension reduction methods such as PCA and UMPA. This reaffirmed the authors' earlier conclusion that different cell types in the VTA, namely dopamine neurons (DA) and GABAergic neurons, exhibit quantitatively distinct input patterns, and a single dose of addictive drugs, such as cocaine and morphine, induced altered labeling patterns. Additionally, the authors illustrate that distinct axes of PCA can discriminate experimental variations, such as minor differences in the injection site of viral tracers, from bona fide alternations in labeling patterns caused by drugs of abuse. While the specific mechanisms underlying altered labeling in most brain regions remain unclear, whether involving synaptic strength, synaptic numbers, pre-synaptic activities, or other factors, the present study underscores the efficacy of an informatics approach in extracting more comprehensive information from the RabV-based circuit mapping data. Moreover, the authors showcased the utility of their previously devised bulk gene expression patterns inferred by the Allen Gene Expression Atlas (AGEA) and "projection portrait" derived from bulk axon mapping data sourced from the Allen Mouse Brain Connectivity Atlas. The utilization of such bulk data rests upon several limitations. For instance, the collection of axon mapping data involves an arbitrary selection of both cell type-specific and non-specific data, which might overlook crucial presynaptic partners, and often includes contamination from neighboring undesired brain regions. Concerns arise regarding the quantitativeness of AGEA, which may also include the potential oversight of key presynaptic partners. Nevertheless, the authors conscientiously acknowledged these potential limitations associated with the dataset. Notably, building on the observation of a positive correlation between the basal expression levels of Ca2+ channels and the extent of drug-induced changes in RabV labeling patterns, the authors conducted a CRISPRi-based knockdown of a single Ca2+ channel gene. This intervention resulted in a reduction of RabV labeling, supporting that the observed gene expression patterns have causality in RabV labeling efficiency. While a more nuanced discussion is necessary for interpreting this result (see below), overall I commend the authors for their efforts to leverage the existing dataset in a more meaningful way. This endeavor has the potential to contribute significantly to our understanding of the mechanisms underlying alterations in RabV labeling induced by drugs of abuse. Finally, drawing upon the aforementioned reanalysis of previous data, the authors underscored that a single administration of ketamine/xylazine anesthesia could induce enduring modifications in RabV labeling patterns for VTA DA neurons, specifically those projecting to the nucleus accumbens and amygdala. Given the potential impact of such alterations on motivational behaviors at a broader level, I fully agree that prudent consideration is warranted when employing ketamine/xylazine for the investigation of motivational behaviors in mice.Specific Points:(1) Beyond advancements in bioinformatics, readers may find it insightful to explore whether the PCA/UMPAbased approach yields novel biological insights. For example, the authors are encouraged to discuss more functional implications of PBN and LH in the context of drugs of abuse, as their labeling abundance could elucidate the PC2 axis in Fig. 2M.

Thank you for this suggestion: we added text (Lines 787-795) discussing the LH and PBN (and GPe) specifically, but also highlighted the importance of our approach in hypothesis-generating science.

(2) While I appreciate the experimental data on Cacna1e knockdown, I am unclear about the rationale behind specifically focusing on Cacna1e. The logic behind the statement, "This means that expression of this gene is not inhibitory towards RABV transmission," is also unclear. Loss-of-function experiments only signify the necessity or permissive functions of a gene. In this context, Cacna1e expression levels are required for efficient RabV labeling, but this neither supports nor excludes the possibility that this gene expression instructively suppresses RabV labeling/transmission, which could be assessed through gain-of-function experiments.

We thank the reviewer for their suggestions regarding this result, and agree that a gain-of-function would be required to provide clearer evidence on this point. We therefore understand that our original phrasing may be misleading. Thus, we have edited this section to the more conservative statement: “These results indicate that reduced levels of Cacna1e likely lower the number of RABV-labeled inputs from the NAcLat, and directly link the levels of Cacna1e and RABV input labeling” (lines 742-744) - we refrain from over-interpreting the results. As mentioned above in response to R1, we added a sentence to explain the rationale behind focusing on Cacna1e (lines 722-724).

**Reviewer #3 (Public Review):**
Summary:Authors mapped monosynaptic inputs to dopamine, GABA, and glutamate neurons in VTA under different anesthesia methods, and under drugs (cocaine, morphine, methamphetamine, amphetamine, nicotine, fluoxetine). They found that input patterns under different conditions are separated, and identified some key brain areas to contribute to such separation. They also searched a database for gene expression patterns that are common across input brain areas with some changes by anesthesia or drug administration.Strengths:The whole-brain approach to address drug effects is appealing and their conclusion is clear. The methodology and motivation are clearly explained.Weaknesses:While gene expression analyses may not be related to their findings on the anatomical effects of drugs, this will be a nice starting point for follow-up studies.

We understand and agree with the suggestion that gene expression allows us to provide correlative observations between in situ hybridization datasets and rabies mapping datasets, and that these results do not show causality. As such, future studies would be needed to assess this in more detail. We have added a line in the discussion to this effect (lines 851-853).

**Recommendations for the authors:**

**Reviewer #1 (Recommendations for the authors):**
Recommendations for improving the writing and presentation:(1) There are a couple of packages available for 3D whole-brain reconstructions based on Allen Brain Atlas (eg. https://github.com/tractatus/wholebrain, https://github.com/lahammond/BrainJ), which would be helpful to align with the gene expression or other data from Allen Institute.

This comment is related to the noted weakness we responded to previously in this rebuttal also from R1 (see comment 1), about the discrepancies between the Franklin-Paxinos atlas and Allen Brain atlas. We agree that a systematic comparison of these two atlases using a tool like wholebrain or BrainJ would be valuable for the field. However, it would be a substantial amount of work, and likely would be an independent study in itself. We believe that the resolution of these atlases was sufficient to make our key conclusions here (e.g., identify gene expression patterns that relate to drug-induced changes rabies virus labeling patterns, and develop a testable hypothesis for CRISPR-based gene editing). They are also based on the same atlases and region definitions that have been applied in our previous studies (e.g., Beier et al., Cell 2015; Beier et al., Nature 2017; Beier et al., Cell Reports 2019; Tian et al., Cell Reports 2022; Tian et al., Neuron 2024; Hubbard et al., Neuropsychophamacology 2025, etc.) The expression of Cacna1e is relatively consistent across the NAc, as we have now detailed in Figure S13.

(2) There are so far two kinds of rabies virus strains available in the neuroscience field (SAD-B19 or CVS-N2c). It is recommended to describe which strain was used in the Material and Methods Section because labeling efficiency and toxicity is quite different between the strains (Reardon TR et al., Neuron 2016).

We have now noted that we used SAD B19 for all experiments (Lines 141-142).

Minor corrections to the text and figures:(1) In Figure 1A, the color differences are not clear (i.e. light gray and dark gray). The figure can be simplified.In addition, generally, images/figures are recommended not to be overlapped with other figures/images (Figures 2A-F, 2G-L).(2) In Figures 7C and D, the authors could add enlarged views of starter cells in VTA and NAcLat.

We have attempted to simplify schematics and figures throughout. High-magnification images of cells have been added as insets in what is now Figure 10 (formerly Figure 7).

**Reviewer #2 (Recommendations For the authors):**
The number of animals for each graph should be explicated within the figure legend. For example, Figure 1C and Figure 7E lack this information. It is also advisable to delineate the definition of error bars within the figure legend.

We have now added mouse numbers to all figures and/or legends, as appropriate. We also indicated in the legend at the end of Figure 1 how error bars and asterisks are defined. Furthermore, we added a sentence to the methods saying that in UMAP and PCA plots each dot is an animal (lines 244-245).

The visual representations, particularly in Figures 1 and 3, are overcrowding. Furthermore, the arrangement of figure subpanels does not consistently adhere to the sequence of explication in the main text, significantly compromising the readability of the text. The authors are encouraged to consider the possibility of segmenting dense figures into two if there exists no upper limit for the number of figure displays. To illustrate, in Figure 3Q, crucial details about experimental conditions are denoted by numerical references, owing to spatial constraints.

We agree that the figure layout and mis-alignment with a linear read of the text was unideal. Therefore, we broke our figures, especially the original Figures 1-4, into multiple sub-figures, including both main and supplemental figures. This facilitated the use of space to rearrange the figure panels, allowing the story to be told in a linear fashion. All figures and panels should now be read in order.

I am seeking clarification on how to interpret the term "overlap" at the bottom of figures illustrating Gene Ontology analysis.

We have clarified the meaning of overlap in this context (lines 324-325): The ‘overlap’ term on the x-axis of these plots means the number of genes in the correlated gene lists that were also within the list of genes for the corresponding GO term.

The authors could provide Cacna1e gene expression patterns within the NAc from the AGEA data.

Cacna1e expression data are now provided in Figure S13.

Additionally, the meaning of "controls" in Figure 7F, along with the "No gRNA" condition, remains ambiguous. While the text mentions "no shRNA", the involvement of shRNA in this experiment lacks clarity.

We now clarify that the control conditions are based on previously published data where no AAVs were injected into NAcLat. This is now clarified in the legend for Figure 10F (lines 1277-1578). We also corrected “shRNA” to “gRNA” in the text.